# Winner–loser plant trait replacements in human-modified tropical forests

Bruno X. Pinho [1,2,3] ✉, Felipe P. L. Melo[3,4], Cajo J. F. ter Braak [5], David Bauman [2,6,7], Isabelle Maréchaux [2], Marcelo Tabarelli[3], Maíra Benchimol [8], Victor Arroyo-Rodriguez [9,10], Bráulio A. Santos [11], Joseph E. Hawes [12,13], Erika Berenguer [7], Joice Ferreira[14], Juliana M. Silveira [15], Carlos A. Peres [13,16], Larissa Rocha-Santos [8], Fernanda C. Souza [17], Thiago Gonçalves-Souza [18,19], Eduardo Mariano-Neto [20], Deborah Faria[8] & Jos Barlow[15] ✉

Anthropogenic landscape modification may lead to the proliferation of a few species and the loss of many. Here we investigate mechanisms and functional consequences of this winner–loser replacement in six human-modified Amazonian and Atlantic Forest regions in Brazil using a causal inference framework. Combining floristic and functional trait data for 1,207 tree species across 271 forest plots, we find that forest loss consistently caused an increased dominance of low-density woods and small seeds dispersed by endozoochory (winner traits) and the loss of distinctive traits, such as extremely dense woods and large seeds dispersed by synzoochory (loser traits). Effects on leaf traits and maximum tree height were rare or inconsistent. The independent causal effects of landscape configuration were rare, but local degradation remained important in multivariate trait-disturbance relationships and exceeded the effects of forest loss in one Amazonian region. Our findings highlight that tropical forest loss and local degradation drive predictable functional changes to remaining tree assemblages and that certain traits are consistently associated with winners and losers across different regional contexts.

Tropical forests are the most important reservoir of terrestrial biodiversity[1] and deliver key ecosystem services for human well-being[2]. Yet, they are being rapidly deforested and fragmented worldwide, with annual losses of 3 to 6 million hectares over the past two decades[3]. This means that a substantial portion of remaining tropical forests exist in human-modified landscapes surrounded by non-forest land uses (for example, pastures, croplands) and exposed to local disturbances such as logging, hunting and fires that lead to degradation[4]. While there is broad consensus that habitat loss is one of the main causes of the contemporary biodiversity crisis[5–7], the independent effects of landscape configuration—here referring to both the number of forest patches and edge density—or local disturbances leading to degradation, remain debated[8–10] or are poorly studied[4]. It is worth noting that most

studies focus on taxonomic diversity, while the mechanisms underlying species loss and gain are underexplored[11]. As plant traits drive species sensitivity to habitat changes and effects on ecosystems functions[12], assessing the specific effects of landscape modification on community functional profiles is critical to improve our understanding and ability to predict biodiversity and ecosystem changes in human-modified tropical forests.

Changes in functional profiles in human-modified tropical forests are unlikely to be simply inferred from changes in species diversity. For example, the population decline of late-successional big trees with high-density woods, large seeds, long life cycles and specialized interactions has been documented in small forest patches and along forest edges, and these 'loser' traits are usually replaced by 'winner' traits

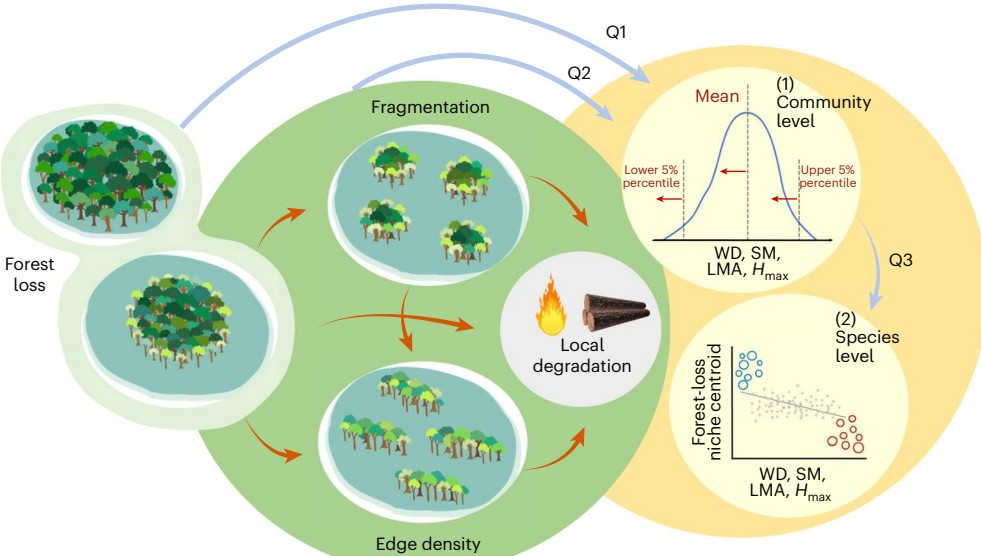

**Fig. 1 | Causal diagram of assumed associations among anthropogenic pressures and the functional outcomes of interest.** The directed acyclic graph (DAG) shows the anthropogenic drivers (green shaded circle) and the outcomes of interest, that is, species-level or community-level functional variables (yellow shaded circle), and the work questions (Q1, Q2 and Q3). Arrows represent the direction of causality between pairs of variables (red and blue arrows for predictor–predictor and predictor–outcome causal associations, respectively). Forest fragmentation (that is, number of patches) typically increases with forest loss, while edge density increases with forest loss and fragmentation level but also with the irregularity of patch shapes. Local forest degradation (for example, timber extraction, hunting, fires) is measured here as the inverse of forest basal area as proxy, and we estimated its independent effects by controlling for the effects of all three landscape-scale disturbance variables. All four disturbance variables are expected to have independent causal effects on the functional profiles of tree communities (in yellow shaded circle, Q2 blue arrow synthesizes three independent arrows from fragmentation, edge density and degradation to the outcome in (1)), driving them toward increased dominance of opportunistic tree strategies characterized by fast growth and high dispersal ability (for example, low-density tissues, small seeds and low-statured trees) and limiting extreme opposite strategies (for example, big trees, large seeds). Also, we expect dominant loser species (red circles) to have contrasting functional profiles from dominant winners (blue circles) ((2) in yellow shaded circle) but that there would be weak overall effects on species distribution (grey line) due to the presence of many species with neutral dynamics or that are regionally rare (grey points). We performed our analyses separately for each of the six regions (Fig. 2), where forest plots are relatively homogeneous in terms of climate and soil types, to limit the risks of potential unobserved confounders of the represented cause–effect relations; this should therefore be read as a 'within-region DAG'. Traits assessed are wood density (WD), seed mass (SM), LMA and maximum tree height ($H_{max}$). Species position on the $y$ axis is defined by the position of their niche centroids in relation to deforestation extent.

typical of pioneer tree species with high dispersal and growth ability[13–15]. This winner–loser replacement may result in weak changes in species diversity but could exert a strong influence on forest biomass, functions and related ecosystem services[16]. In addition, the literature on taxonomic shifts does show the importance of separating out patch-level observations from the overarching role of forest loss. For example, on the one hand, there is evidence of reduced species diversity in small, isolated, edge-affected forest patches, driven by dispersal limitation, edge effects (that is, physical and biotic changes at the margins of forest fragments) and demographic stochasticity[6,15]. On the other hand, empirical studies and meta-analyses over the past decade also show that patch-level ecosystem decay does not necessarily imply negative effects of landscape-scale forest fragmentation, as for a given forest cover, landscape configuration appears to have weak effects on species diversity at both local and landscape scales[7,9,17]. Some of the apparent contradictions can be explained by the limitations of using species diversity as a response metric[10], and it is still unclear whether the taxonomic rearrangements of tropical forest tree communities can be translated into generalizable and predictable changes in functional traits that describe the whole spectrum of plant strategies in the world[18].

Where studies have addressed the functional rearrangement of biodiversity in human-modified tropical landscapes, the advances have been far from complete[19]. First, studies on landscape-scale predictors of functional profiles of tree communities have been restricted to a few regions, making it difficult to tease apart context-dependent patterns due to particular land-use and biogeographic histories from widespread patterns resulting from a consistent imprint of landscape modification[20]. Second, the available evidence is biased to a few functional groups[13,21] or

plant traits, such as wood density[22] and dispersal/pollination modes[23,24], although ref. 25 assessed more comprehensive functional profiles. Finally, these few functional studies have mainly addressed changes in community trait mean and functional diversity metrics. Although valuable, these approaches overlook other important functional aspects of communities, such as extreme trait values (for example, very big trees and large seeds) that have proven to drive ecosystems[26,27]. Furthermore, changes in trait mean values are usually interpreted as a consistent trait-mediated species sorting along disturbance gradients, when it could represent the replacement of a few dominant winner–loser species with contrasting functional profiles[28], although this is still a hypothesis to be tested[19].

In this Article, we address these knowledge gaps by applying a causal inference analytical framework (Fig. 1). Using a large dataset of tree abundance and traits across six Neotropical rainforest regions with different patterns and histories of land-use change (Fig. 2), we ask whether there is a consistent trait-based outcome of landscape modification and local degradation. Specifically, we examine the following: (1) which traits are most sensitive to landscape-scale forest loss, (2) whether landscape configuration and local degradation cause functional changes beyond the effect of forest loss and (3) whether community-level functional changes reflect consistent trait-mediated species sorting along disturbance gradients or rather the replacement of a few winner–loser species. These questions were underpinned by the following predictions (Fig. 1, yellow shaded circle). We expected that forest loss increases the dominance of disturbance-adapted species with fast resource-acquisition traits (that is, low-density leaf and wood tissues), low stature and high colonization ability (that is, small

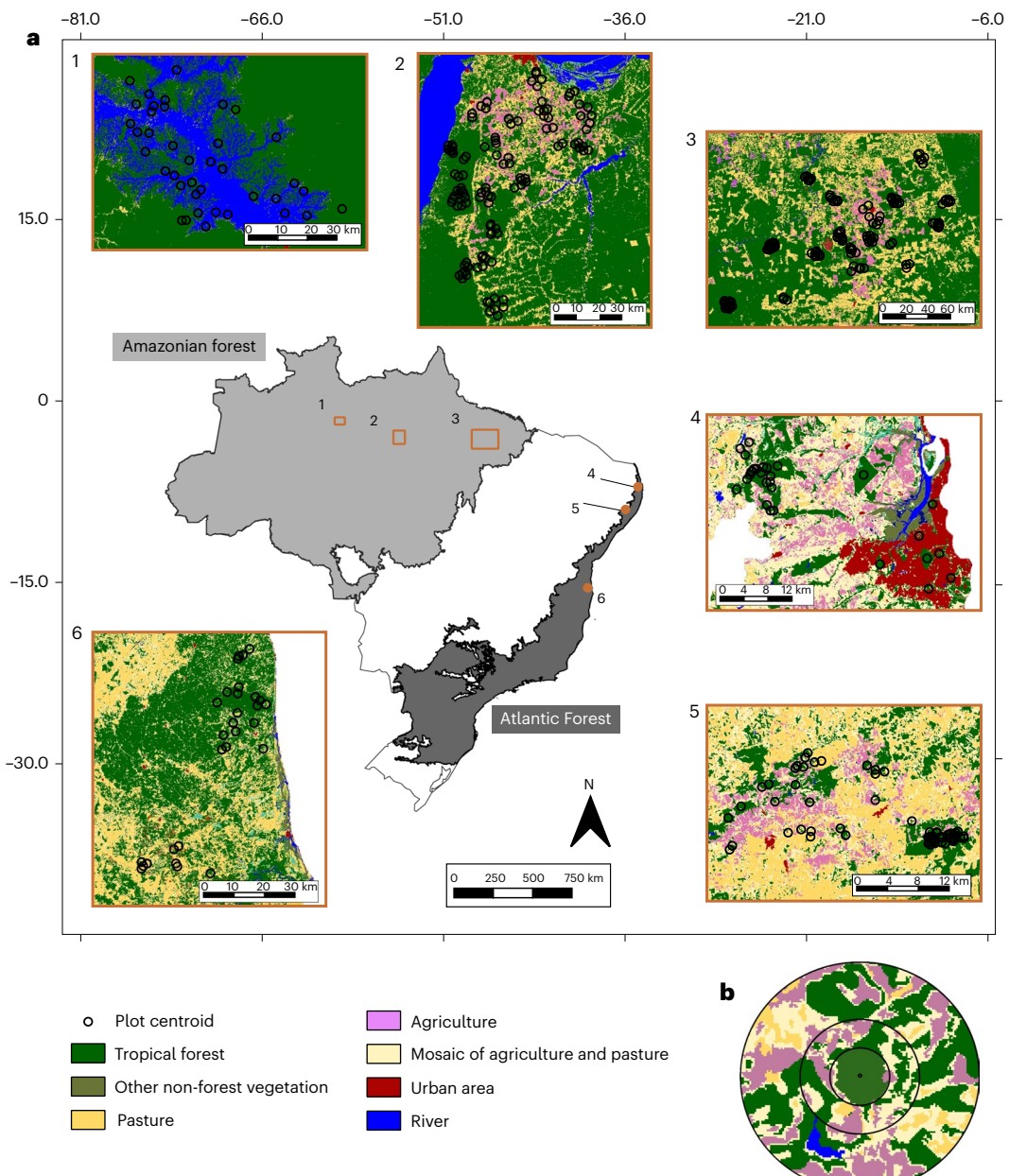

**Fig. 2 | Location of the six study regions and study plots across the Amazonian and Atlantic forests in Brazil. a**, Black circles illustrate the landscapes surrounding each study plot at the largest analysed spatial scale. **b**, Our multiscale approach showing concentric buffers (that is, 'local landscapes'; ref. 5) surrounding each plot (500, 1,000 and 2,000 m radii), within which we assessed the landscape variables. The study regions (1, Balbina; 2, Santarém; 3, Paragominas; 4, Paraíba; 5, Serra Grande; 6, Una) have markedly different land-use patterns and histories, as they include some of the oldest agricultural frontiers in Brazil (regions 4 and 5) and regions recently disturbed by flooding for energy generation (region 1) or for agriculture (regions 2 and 3). They also have different anthropogenic matrices, including water (region 1), diverse agriculture (regions 2 and 3), sugar cane monoculture (region 4), urban areas (region 5) and mixed agroforestry and pasture (region 6).

seeds dispersed by either abiotic factors or endozoochory) and shifts the limits of the trait distributions by limiting extreme opposite trait values (that is, big trees, conservative resource use, large seeds dispersed by synzoochory) and extending them towards more extremely acquisitive and small-seeded trees (Q1, Fig. 1, (1) in yellow shaded circle). Regarding our second aim, we expected that the independent effects of landscape configuration and disturbances leading to degradation can exacerbate the negative effects of forest loss (Q2), as they can favour disturbance-adapted species and discriminate against conservative resource-use strategies[29] and big trees[27]. Finally, we expected changes in community functional profiles to reflect the replacement of a few dominant winner–loser tree species with contrasting functional

profiles, rather than a consistent response across species (Q3, Fig. 1, (2) in yellow shaded circle).

We adopt a structural causal modelling framework[30,31] to estimate the total causal effects of each disturbance variable according to our assumptions of causal relationships among them and with the outcome variables (Fig. 1). In this framework, we assume from well-documented relationships that forest loss leads to increased number of forest patches (that is, fragmentation), both of which increase forest edge density[8,32], and that these three landscape-scale drivers lead to increased local degradation[4]. To account for these multiple cause–effect relations, we used separate models with different sets of control variables aimed at estimating the causal effects of each driver

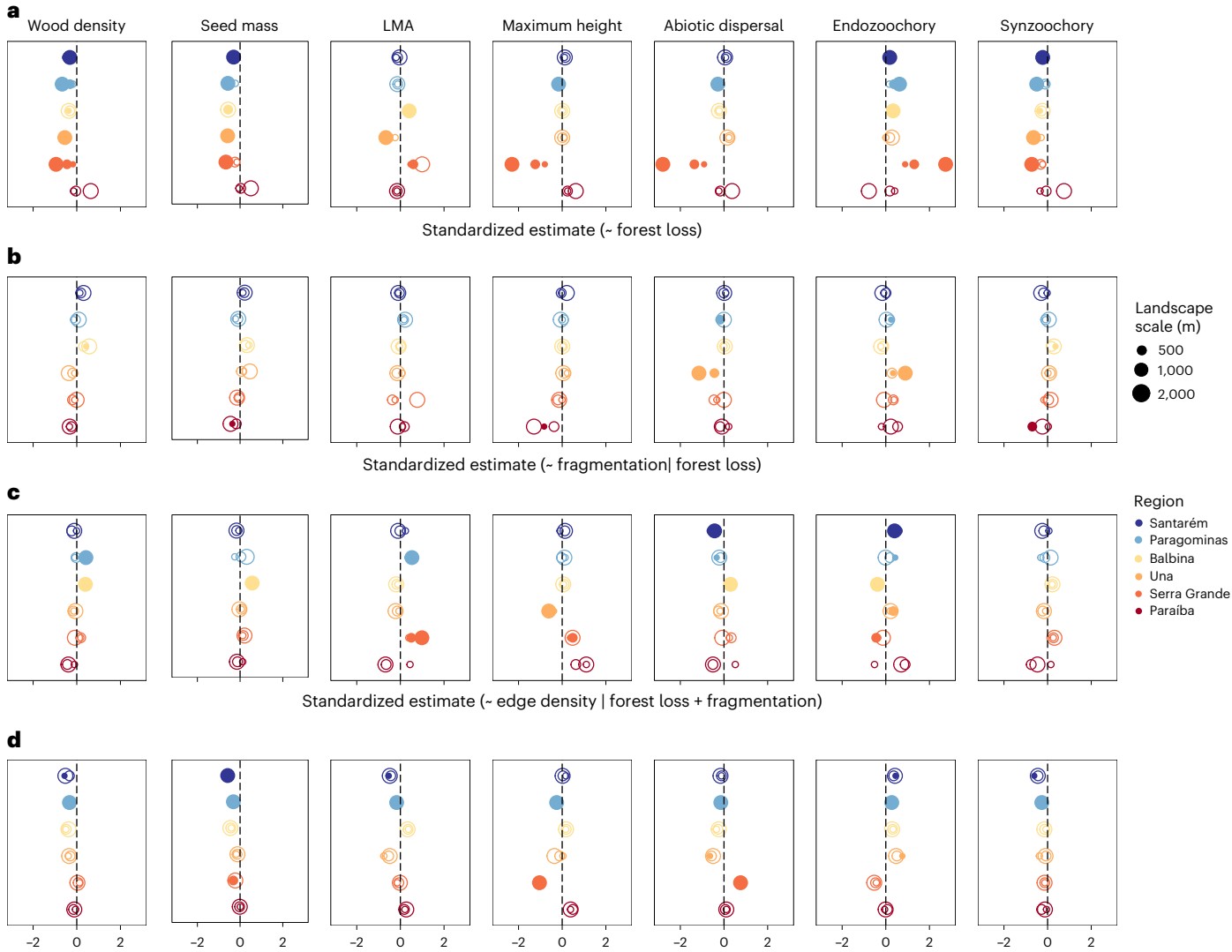

**Fig. 3 | Direction and magnitude of the responses of abundance-weighted community trait means to anthropogenic drivers of change. a–d**, Points represent the standardized causal effect estimates (that is, effect sizes) of models used to estimate the total causal effects of landscape-scale forest loss (**a**), and the independent effects of forest fragmentation (that is, number of forest patches) (**b**), edge density (**c**) and local degradation (**d**) across 271 old-growth forest plots in six Neotropical regions (colours) across the Amazonian and Atlantic Forest biomes in Brazil (Fig. 2a). Landscape drivers were measured across multiple spatial scales (circle sizes) surrounding each sample plot (Fig. 2b), with filled circles representing the scales at which we found significant responses (that

is, those estimates with 95% confidence intervals non-overlapping the zero effect). Predictors and traits were standardized, so that comparing causal effect estimates among the four models reflects relative trait responses to increases of one standard deviation in the predictor of interest. Models to estimate the causal effects of each driver have specific sets of control variables, which are shown after the vertical bars in the x axis, defined to close all non-causal paths leading to the response variable (Fig. 1; details in Methods). Regions are ordered from the least to the most disturbed (top–down) according to overall forest cover and land-use history (Extended Data Table 1).

on the outcomes of interest (Fig. 1; details in Methods). As all regions were originally covered by tropical forests, we measured forest loss as the percentage of the landscape covered by non-forest land covers, including cattle pastures, agricultural lands and human settlements. This percentage thus represents the cumulative forest loss to date. We also measured the number of forest patches and forest edge density in each landscape. As we do not know a priori the scale of landscape effects[33], we measured each of these three landscape variables at three spatial scales, using concentric buffers of 500, 1,000 and 2,000 m radius around each forest plot (Fig. 2b). To quantify local forest degradation, we measured the inverse of tree community basal area, a proxy of forest biomass that is negatively affected by local disturbances[34,35].

At the community level, we analysed changes in community-weighted means (CWM) and 5% percentiles (lower and upper) of trait distributions,

the first describing trait dominance and the latter extreme trait values in a community (Fig. 1, (1) in yellow shaded circle). Species distributions along forest loss gradients were parameterized by species niche centroids, calculated as the abundance-weighted average forest loss across the plots where a species occurs. Winners and losers were defined as those species whose distribution differs from a random-dispersal null model[36], in which winners thrive with forest loss in opposition to losers whose abundance is reduced in more deforested landscapes. We then assessed functional differences among winners and losers and tested trait effects on species niche centroids (Fig. 1, (2) in yellow shaded circle). Community and species-level models were further fitted separately for each trait in each region at each landscape scale. Finally, we tested the robustness of our findings by applying a double-constrained correspondence analysis (dc-CA[37]) to all vegetation, trait and environment data simultaneously

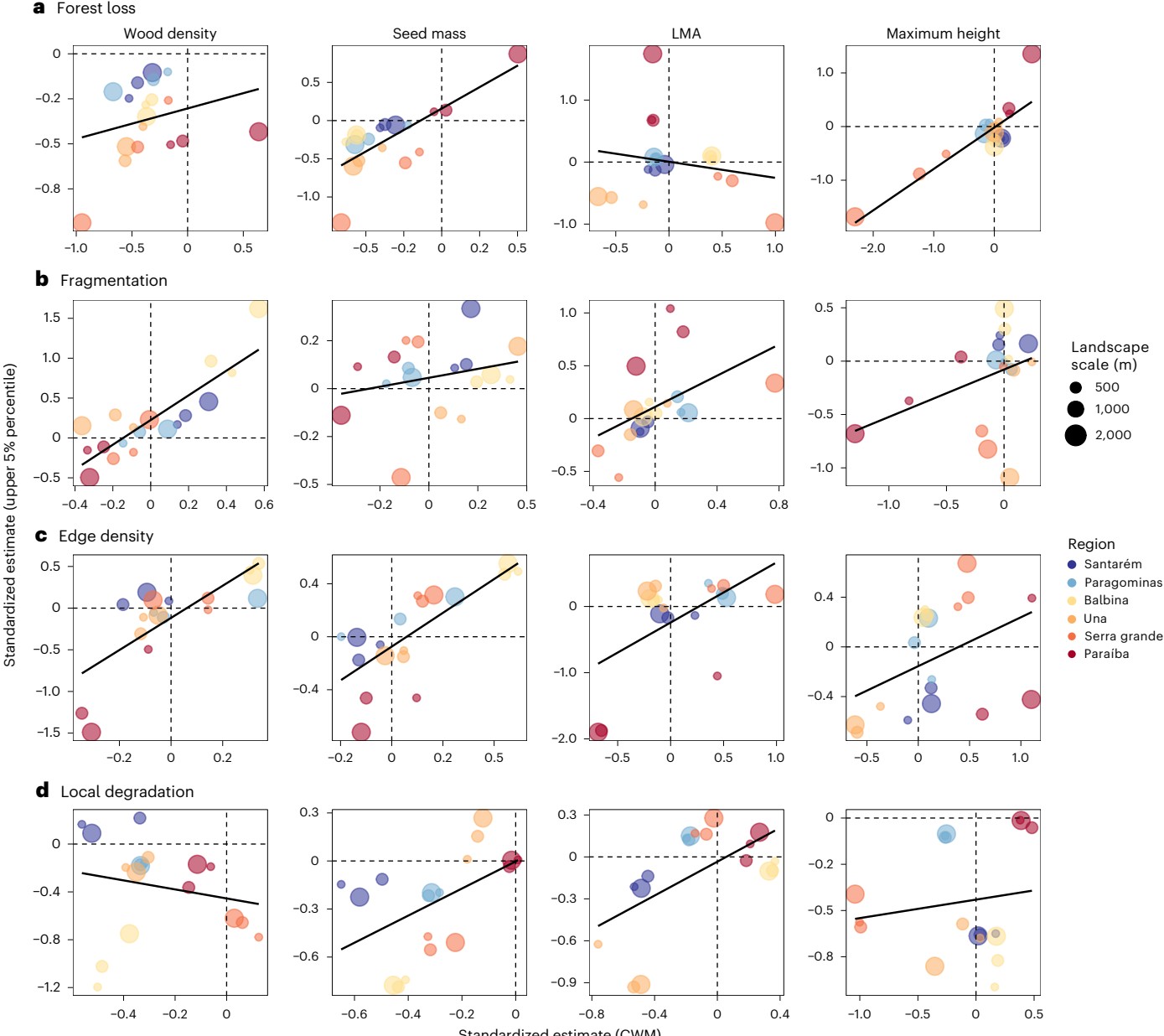

**Fig. 4 | Correlations between effects on abundance-weighted means and upper 5% percentiles of community trait distributions. a–d,** Points represent the standardized causal effect estimates (that is, effect sizes) of models, colours represent the study regions and point sizes represent the spatial scale at which we measured the landscape variables. Traits are organized by columns and predictors by rows. To estimate the total causal effect of forest loss (**a**), fragmentation (**b**), edge density (**c**) and local degradation (**d**), we used models with specific sets of control variables (see Fig. 3). Points in the bottom left and upper right sides of each panel (separated by dashed lines showing zero effects) illustrate similar effects in direction, while those in the bottom right and upper left sides illustrate contrasting effects. Correlations with lower 5% percentiles are shown in Extended Data Fig. 4.

## Results

### Forest loss causal effects on community functional profiles

The percentage of forest loss in the surrounding landscapes affected CWMs of all analysed traits in at least two and up to five of the six study regions; only Paraíba, which has the longest history of disturbance and greatest deforestation level, did not show any significant relationships (Fig. 3a). Overall, when significant, forest loss effects

using the above analytical framework. The dc-CA combines multivariate CWM disturbance and species niche centroids-trait regressions in an eigen analysis to find the best composite trait and disturbance gradient, guarding against type 1 error rate inflation in CWM regression[28]. See Methods for further details on our analytical approach.

were consistent across regions and spatial scales and in accordance with our predictions, for all traits except leaf mass per area (LMA) and abiotic dispersal. For instance, wood density, seed mass and syn-zoochory were negatively associated with the percentage of forest loss in all but one region (Fig. 3a). The total causal effect of forest loss alone explained an average 20% (range of $R^2_{adj.}$ = 9–55% range) of variation in CWM trait values across trait–region combinations (Extended Data Fig. 1).

Forest loss shifted both extremes of the trait distributions (that is, lower and upper 5% percentiles) towards lower values in half of the trait–region combinations (12/24) (Extended Data Figs. 2 and 3). With the exception of LMA, these effects were positively correlated with the observed effects on CWMs (Fig. 4a and Extended Data Fig. 4a). However,

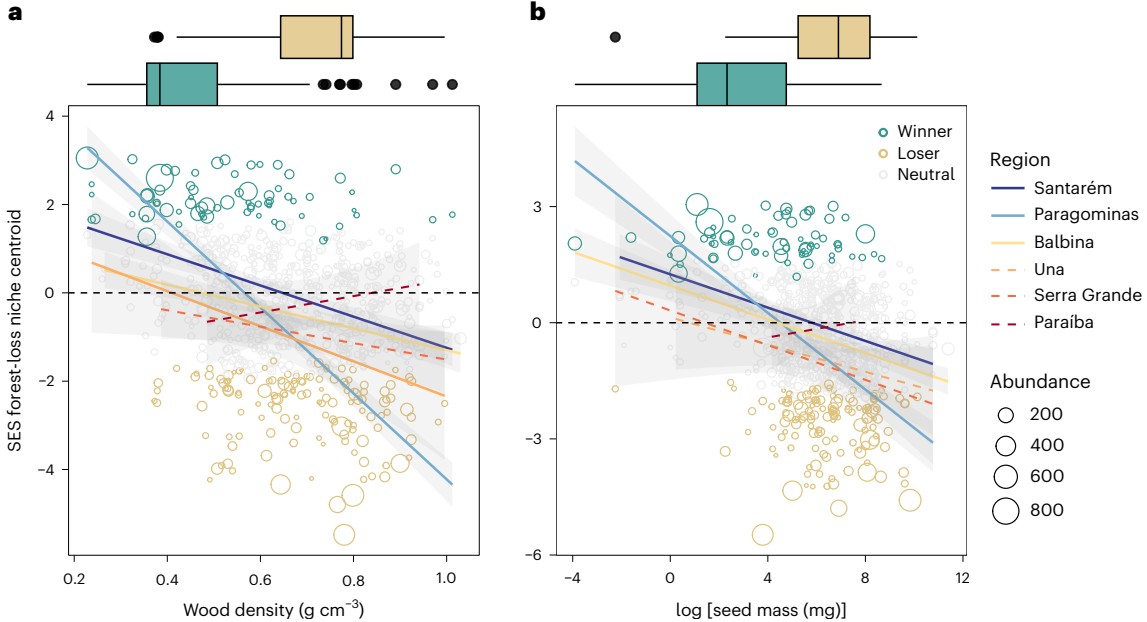

**Fig. 5 | Winner–loser species trait replacements along landscape-scale gradients of forest loss. a,b,** We show the results at the largest analysed spatial scale (2,000 m radius buffers) in six Neotropical forest regions (line colours), for the two traits with stronger and more consistent responses at the community level (Fig. 3): wood density (**a**) and seed mass (**b**). We consider winner or loser species as those whose abundance-weighted distribution (that is, niche centroid) along forest loss gradients significantly deviate positively (winner, green circles and boxplots) or negatively (loser, yellow circles and boxplots) from a random-dispersal expectation, respectively, while neutral species (beige points) are those non-significantly related to forest loss. Model-fit lines illustrate the linear relationships between tree species' niche centroids and their functional traits (*n* = 765 species–region combinations for 484 species occurring in at least five

plots in a region). Both model-fit lines and boxplots illustrating winner–loser trait differences are weighted by species total abundance across plots for each species–region combination, which performed better than non-weighted analyses. In the scatterplots, significant trait effects are illustrated by solid lines and non-significant effects by dashed lines, with shaded areas indicating 95% confidence intervals. Boxplots indicate the median (centre line), 25–75% quartiles (box edges), <1.5 times the interquartile range (whiskers) and extreme values (dots). Black dashed lines in the zero *y*-intercepts separate species with positive and negative deviation of observed niche centroid from random expectation (that is, standardized effect sizes, SES). In addition, we show the relative position of winners and losers of forest loss in a multivariate principal component analysis functional space in Extended Data Fig. 6b.

they were weaker and less predictable compared with the observed effects on CWMs, with an average 15% (5–25%) and 26% (11–44%) of explained variation for upper and lower 5% percentiles, respectively.

**Fragmentation, edge density and degradation causal effects**
From the different models focused on the causal effect of each of the anthropogenic drivers of trait change, we found that the independent effects of landscape configuration (that is, number of patches, edge density) and local degradation on CWM traits were either relatively rare, weak or inconsistent across traits and regions (Fig. 3b–d). In particular, the number of forest patches showed a weak effect in most regions, being significant in 0 to 2 regions per trait. Yet, this landscape feature was relatively important in Paraíba, where it negatively affected seed mass, maximum height and synzoochory (Fig. 3b). The causal effect of edge density was significant for some traits and regions, and, contrary to our expectations, the direction of effects was mostly positive for traits such as wood density, seed mass and LMA (Fig. 3c). Finally, causal effects of local degradation were evident in 1 to 3 regions per trait, and when significant, they supported our predictions, leading to decreases in wood density, seed mass, LMA, maximum height and synzoochory (Fig. 3d). These effects were particularly strong in the Paragominas region, where the independent effect sizes for local degradation often exceeded those resulting from forest loss (Fig. 3d). While different models were necessary to assess the respective causal effects of each disturbance variable (Methods), adding the number of patches, edge density and local degradation in a separate analysis aimed at optimizing out-of-sample expected predictive accuracy improved the explanatory power of the forest loss-only model to an average of 32% (8–63%) across all CWM traits and study regions.

The independent causal effects of these landscape configuration and local degradation drivers on extreme trait values (Extended Data Figs. 2 and 3) were positively correlated to those on CWM traits for all but two trait–driver combinations (Fig. 4b–d and Extended Data Fig. 4b–d). However, these relationships were much weaker and less predictable, with combinations of the four predictors explaining an average of 17% (2–40%) and 28% (6–59%) of the upper and lower 5% percentiles, respectively, across trait–region-scale combinations.

**Winner–loser species replacements**
We identified winner and loser species along forest loss gradients in all regions but Paraíba, where there were no winners and only a few losers (Fig. 5 and Extended Data Fig. 5a). However, the distribution of most species was not caused by the percentage of forest loss in the landscape ('neutral status' in Fig. 5), with an average 8% of winners (2–21%) and 20% of losers (9–39%) across regions (Extended Data Fig. 5a). Overall, species traits were significantly and negatively related to forest-loss species niche centroids in four out of the six study regions for wood density, and three out of six for seed mass (Fig. 5), explaining an average 5% (2–17% across trait–region combinations) of variation in species distributions along forest loss gradients. Weighting by species abundances improved model performance and predictive ability, with an average 17% and up to 53% of explained variation across trait–region combinations. Despite these overall weak trait–niche relationships, winner and loser species were clearly distinguished functionally, with losers bearing harder woods and larger seeds (boxplots in Fig. 5 and Extended Data Fig. 5 for a more complete analysis of functional differences among winners and losers). These differences were more pronounced when weighted by species abundances.

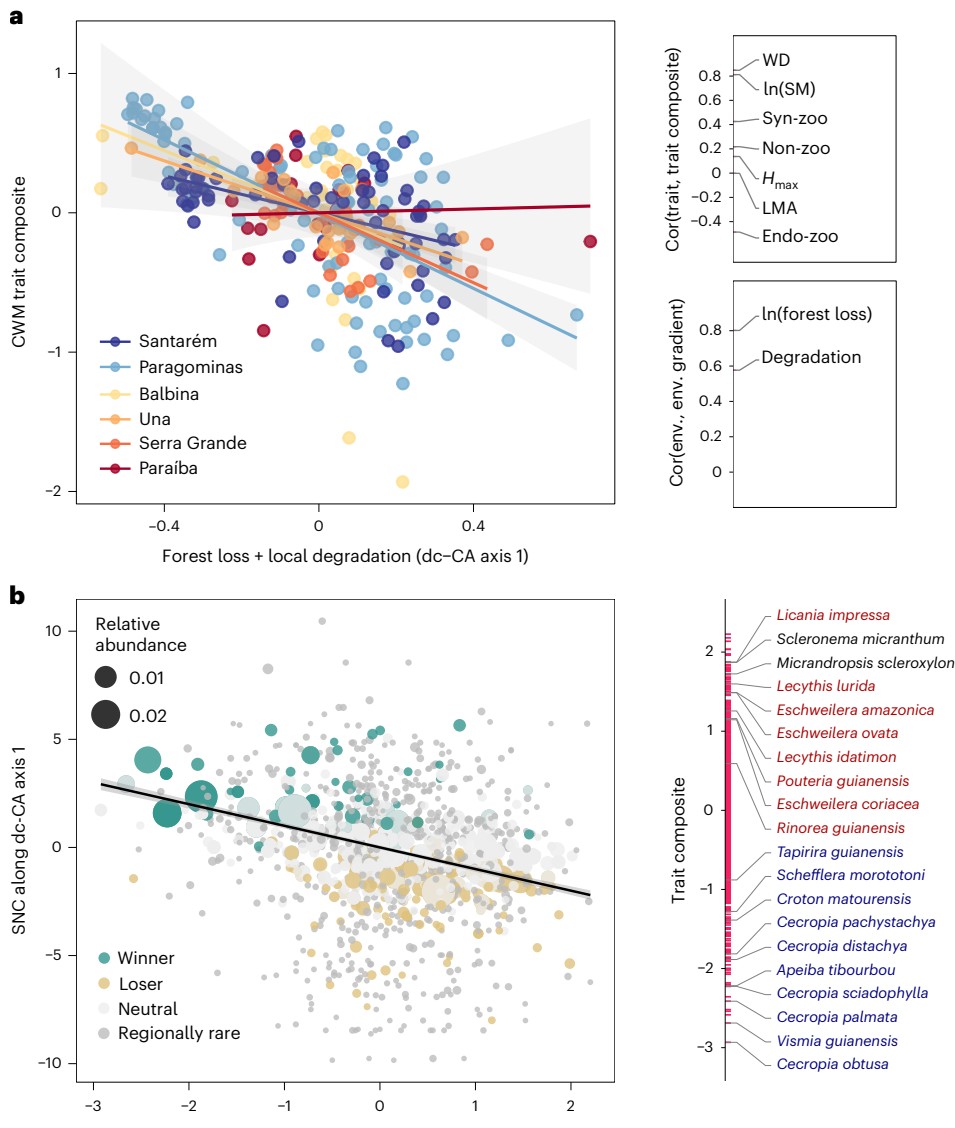

**Fig. 6 | Multivariate trait-disturbance relationships for 271 tree assemblages and 1,207 species across six Neotropical forest regions. a**, Region-specific relationships between the first axis of a dc-CA, representing forest loss and degradation (at 2,000 m scale), and the abundance-weighted mean of the best trait composite at the assemblage level. Side plots show correlation coefficients between each functional trait and the best trait composite (upper plot), as well as between each environmental variable (env.) and the disturbance axis (lower plot). **b**, Species-level assessment of the relation between the trait composite and species niche centroids (SNC) along the axis of forest loss and local degradation (dc-CA axis 1). The side plot shows the trait composite values of the 20 species with highest contribution to the dc-CA axis 1, of which 10 are winners (in blue) and 8 are losers (in red). Shaded areas around model-fit lines in **a** and **b** indicate 95% confidence intervals. Syn-zoo, synzoochorous; Endo-zoo, endozoochory; Non-zoo, abiotically dispersed.

## Multivariate trait-disturbance relationships

Using dc-CA, all three landscape variables at each of the three analysed scales and local degradation were significant ($P < 0.01$, even after the conservative Bonferroni adjustment), with forest loss at 2,000 m scale being almost twice as important as the next best variable, local degradation. However, the independent effects of forest fragmentation and edge density (that is, after accounting for forest loss and forest loss with fragmentation, respectively) were not significant, suggesting that the multivariate trait composition of tree assemblages is weakly related to landscape configuration. This result supports the findings from the univariate causal analytical framework (Fig. 3). Local degradation continued to be significant ($P < 0.01$) even after accounting for forest loss, fragmentation and edge density. In the dc-CA with the two significant predictors (forest loss and local degradation) and all traits, only the first axis was significant ($P = 0.001$). We interpret this axis as a composite disturbance gradient with forest loss and local degradation

contributing in a ratio of about 2 to 1 in terms of standardized weights in the regression against the CWM trait composite with the covariate region. All traits were positively correlated with the trait composite gradient, except endozoochory (Fig. 6a). The largest two weights in the trait composite were those of wood density and seed mass—which again supports their strong responses in the univariate causal analytical framework. The CWMs of the trait composite decreased along the composite disturbance gradient (that is, dc-CA axis 1) (Fig. 6a), and the species niche centroids along this gradient decreased with the trait composite (Fig. 6b).

## Discussion

Our results suggest that landscape-scale forest loss and local degradation drive consistent changes in the functional profiles of local tree communities across human-modified neotropical forests in different biogeographic, climatic and land-use contexts. We focus here on four

contributions emerging from the work: (1) the prominence and consistency of forest loss effects versus other anthropogenic drivers, (2) the variation in the responses across different traits, (3) the association of changes in community trait means with the loss of extreme trait values and (4) the role of a few dominant winner–loser species with contrasting traits as drivers of changes in community functional profiles.

The strong overarching influence of forest loss on the functional profiles of tree communities supports our expectation that part of the effects of other drivers are mechanistically associated with (and therefore can be predicted by) forest loss (Fig. 1, green shaded circle). For instance, the negative effects of forest loss often capture those of fragmentation and edge density, as increasing forest loss usually implies increased fragmentation and therefore edge density[8,32]. Beyond capturing landscape configuration effects, Fahrig's habitat amount hypothesis[33] suggests that the amount of habitat in surrounding landscapes is an important determinant of local species diversity, as more habitat equates to more individuals and species that could potentially colonize a local community. It remains to be seen how these potential mechanisms related to diversity could relate to functional profiles, but it is reasonable to expect that forest loss particularly limits the arrival of species with lower fecundity and dispersal capacity, which may explain the predominance of opportunistic species with high fecundity and dispersal capacity (small and endozoochoric seeds) in the more deforested landscapes we examined.

It was also clear that forest loss is not the only driver of change, although the strength of additional effects of other drivers was often context dependent. For instance, the important influence of local disturbances in Paragominas likely reflect the region's history of conventional logging and fires[34,38]. These disturbances are not always associated with remaining forest cover, and their strong effects show the importance of policy interventions that go beyond avoiding deforestation[4,29]. Also, negative effects of fragmentation per se were often only significant in Paraíba, which has the greatest regional level of deforestation. This may suggest that, as the 'fragmentation threshold hypothesis' postulates[39], negative effects of fragmentation are stronger in more deforested regions[25] due to increased prevalence of isolated and edge-affected forest patches. Finally, the inconsistent but often significant trait-level responses to edge density also require further research but may reflect the complexity of abiotic and biotic edge effects that are highly variable in space and time as they depend on multiple factors, such as matrix type, edge age and orientation, and interactions among nearby edges[40–42]. Indeed, the 'landscape-divergence hypothesis' suggests that edge effects are spatially variable and temporally dynamic[41], which can result in contrasting successional trajectories in different regions[43].

The use of a causal framework allowed us to infer total causal effects of our disturbance variables while minimizing risks of over-control, confounding or collider biases[30,31]. However, the inferences remain conditional on the accuracy of the causal model (Fig. 1) and the presence and frequency of different condition combinations in the available data (for example, low forest loss, high fragmentation). Forest loss is an economic and social process that is likely to limit the empirical data to a subset of potential combinations. Going beyond this will require experimental landscapes (for example, Stability of Altered Forest Ecosystem - SAFE Project, www.safeproject.net) or the use of mechanistic simulations (for example, ref. 44) that allow wider and orthogonal gradients of forest loss and fragmentation. Furthermore, we were limited by relying on snapshots of tree communities at specific times in each region. Understanding the temporal dynamics of human modified forests will help reveal the time lags between disturbances and functional changes (that is, the functional extinction debt) and how tree species and traits are responding to climate change and its interactions with local anthropogenic pressures.

Overall, human modification brings about an increased dominance of 'opportunistic species' that can be defined here as having fast

growth (low-density woods), high fecundity and high dispersal ability (small, endozoochorous seeds consumed by mobile frugivorous vertebrates). Some of these changes reflect known trait-mediated vulnerabilities. One of the most pervasive changes is the loss of large-seeded tree species, whose combined dependence on large-bodied seed dispersers and physiological requirements for germination make them especially vulnerable to both local and landscape-scale disturbances[24,26,45]. Although the association between seed size and wood density is weak at the species level, it can be strongly expressed at the community level due to the dominance of certain small-seeded and low-wood density species dispersed by the many disturbance-adapted and highly mobile bats and birds that proliferate in human-modified landscapes[24,26,46].

Some results go against the prevailing understanding of trait responses to human disturbance. For example, we expected wood density to be negatively associated with edge density but observed either no independent effect or an increased dominance of hard-wooded species in response to edge density in some Amazonian regions. While fast-growing pioneer trees may benefit from increased light availability in forest edges[13,15], the harsh conditions of desiccated soil and air (that is, water stress)[40], frequent fires and sporadic windstorms could also favour hard-wooded trees with conservative resource-use traits that make them tolerant to abiotic stresses and resistant to stem snapping and uprooting[47,48]. This finding highlights the context dependency of responses of tropical flora to disturbance regimes and shows the need for more research that can understand and unpick the role of land-use history and biogeography[20,49], including the use of physiological traits that provide direct mechanistic links between environmental drivers and plant responses[50].

The loss of abiotically dispersed trees is somewhat more surprising given the prevalence of wind dispersal among ruderal species in other biomes; although this may reflect the failure of this non-directional dispersal mode in fragmented landscapes[51], where seeds most often end up in a hostile matrix, this was not supported by the inconsistent or non-significant independent effects of fragmentation (Fig. 3). The rarity of significant effects on maximum height can be related to the existence of some exceptionally tall long-lived pioneers, as evidenced by the orthogonal variation of this trait in relation to the resource-use traits (for example, wood density, seed mass) that distinguish winners and losers (Extended Data Fig. 6b). Finally, the highly inconsistent response of LMA is likely because it responds to multiple environmental gradients (for example, light, nutrient availability)[52]. Furthermore, the functionality of traits—whether they affect individuals' vital rates and fitness—is dependent on the abiotic and biotic conditions[53], such that LMA may not have particularly high adaptive value along the studied gradients or have complex adaptive values (for example, age- or size-dependent interactions with other traits or environmental conditions) that our analytical framework did not capture.

In recent years, there has been growing awareness that there is no single measure of community-level trait distributions and that CWM can—when taken alone—be problematic, by returning type 1 errors[28,54], masking effects at the edge of the trait spectrum or by amalgamating contrasting trait strategies[55]. We shed light on this important issue in three different ways. First, we confirmed the robustness of our findings from the causal analytical approach, by summarizing both the community- and species-level responses to anthropogenic change drivers (Figs. 3 and 5) using dc-CA. A dc-CA with forward selection of disturbance variables at the three spatial scales led to the same final model as the directed acyclic graph (DAG)-based dc-CA modelling approach but would have had less power and would not have allowed causal inference. Second, we show that effects on CWM and extreme trait values were mostly correlated, and the latter mostly held similar—although weaker—relationships with the anthropogenic drivers. This suggests that these drivers usually push the whole trait distributions toward lower values—that is, increasing dominance of more opportunistic strategies but also limiting extremely conservative resource-use

strategies and allowing the presence of extremely acquisitive strategies that are absent from more conserved landscapes. However, this was not always the case, and the number of weak and unclear relationships also suggest extreme trait values can respond differently to CWM traits.

Finally, we found that although winners and losers were functionally distinct, they were not distinguished from species irresponsive to forest loss gradients, and both winners and losers covered large ranges of trait values (boxplots in Fig. 5 and Extended Data Fig. 5a). However, a few dominant winners and losers (large green and yellow points in Fig. 5) were consistently found at the extremes of the traits' distribution, corresponding to either low or high trait values, respectively. This suggests that while many trait dimensions may ultimately determine the fate of species in human-modified landscapes, dominant winner and loser species have stronger and more consistent trait–environment associations—those embedded in landscapes with high forest cover consistently held extremely dense woods and large seeds dispersed by synzoochory (loser traits), while species dominant in highly deforested landscapes had low-density woods and small seeds dispersed by endozoochory (winner traits).

The changes in functional strategies of dominant species we document here could have important consequences for ecosystem functioning. The proliferation of disturbance-adapted, fast-growing, small-seeded endozoochorous tree species usually correlates with declines in forest productivity and carbon storage potential[16,26,35], while the loss of large-seeded and synzoochorous tree species will change fauna–flora interactions and influence long-term regeneration potential[56,57]. Recent research has shown hyperdominance is prevalent across the world's primary tropical forests[58]; our work suggests hyperdominance occurs in human-modified forests, too, and that these dominant species define the traits—and therefore functioning—of these forests.

From a policy perspective, our findings reinforce the need to (1) preserve and restore as much forest as possible[59], (2) limit degradation of remaining forests[4] and (3) conserve or restore vulnerable trait combinations, such as large-seeded and high wood-density species. As many of these vulnerable trait combinations involve synzoochory, this will require actions that support their dispersal agents, such as by reducing hunting or supporting reintroductions of large-bodied birds and mammals.

## Methods

### Study regions

We studied 271 old-growth forest plots distributed across six Neotropical regions, three located in central eastern Amazon (Paragominas, Santarém and Balbina) and three in the northeastern Atlantic forests (Una, Serra Grande and Paraíba) in Brazil. The study regions cover most of the latitudinal distribution of neotropical forests in the southern hemisphere (Fig. 2a) and encompass a broad range of climate and land-use histories (Extended Data Table 1) including different matrix types and hostilities, times since deforestation and logging pressures within the remaining forests. The three Amazonian regions have a relatively recent (<60 years) history of extensive land use change. The municipality of Paragominas is the most recently occupied, and lost around 40% of forest cover since the construction of the Belém–Brasília highway and founding of the city in 1961. The Santarém region encompasses multiple municipalities on the east bank of the Tapajós river; this has a much longer history of human occupation, but much of the deforestation in the wider region accelerated over the past 30 years with the development of soybean export terminus and the paving of the Santarém–Cuiabá highway. Both regions are dominated by a mosaic of agricultural lands and pastures, with some timber plantations and smallholdings[22,34]. Balbina is an 'archipelago' of small forest islands completely surrounded by a large body of freshwater due to the flooding of a reservoir lake after the closure of the Balbina Hydroelectric Dam in 1986[60]. We assess three regions in the Atlantic Forest, which

is a global biodiversity hot spot[61] with about 20% of its original forest cover remaining, 97% of the forest fragments being smaller than 50 ha[62]. These regions have much longer histories of large-scale deforestation (300–500 years), with some dating back to the Portuguese arrival in Brazil. The Una region represents a hot point within the Atlantic Forest hot spot[63], retaining high levels of tree species endemism and biodiversity. Large-scale deforestation in this region only started in the 1960s and intensified in the 1990s with the cocoa economic crisis. The matrix in the region is a mosaic composed mainly of cocoa agroforestry, rubber and eucalyptus plantations. The Serra Grande region consists of one relatively large and several small forest patches surrounded by sugar cane crops that are burnt annually for harvesting, leading to strong edge effects[13]. Finally, plots in the Paraíba State represent the most disturbed scenario, with forest patches mostly surrounded by urban areas in the capital João Pessoa[64] and sugar cane plantations in surrounding municipalities[65].

### Vegetation data

We used data from 28,565 adult trees (with diameter at breast height, DBH ≥ 10 cm, excluding lianas and palms) of 1,207 species belonging to 76 botanical families. The selection of adult trees is a conservative approach as landscape modification effects may be stronger and should manifest earlier in seedlings and saplings[22,25]. The methods adopted for vegetation inventories are described elsewhere[13,22,23,34,60,64–66]. Plot sizes vary among regions but differ only slightly within regions (that is, the scale of our analyses) in the Atlantic Forest (Extended Data Table 1). This should not affect our results because we did not focus on diversity measures or changes across regions but rather on relative abundance-weighted measures, compared among communities within regions. The Amazonian and Atlantic forests have clearly distinguished floras, but there was some overlap in species composition among regions within these biomes (Extended Data Fig. 7).

### Functional traits

We compiled data for five species traits related to plant resource use and regeneration in all plant organs: wood density (g cm$^{-3}$), seed mass (mg), LMA (g m$^{-2}$), maximum tree height (m) and dispersal syndrome (abiotic dispersal, endozoochory, synzoochory). These were compiled from previous publications by the authors[22,24,49,64,65] and the plant trait databases TRY[67], Botanical Information and Ecology Network[68] and Seed Information Database[69].

These traits position species along the plant economics and size-related trait spectra[18] and are known to affect plant performance along stress and disturbance gradients[12,70]. The leaf and stem traits considered are related to a trade-off between fast resource acquisition and growth in resource-rich/disturbed environments, and slow growth but high survival under abiotic stress by investment in dense, well-protected tissues[12]. Seed mass is related to species' regenerative strategies and reflect a tolerance–fecundity trade-off, in which large-seeded species are more tolerant to stressful conditions (for example, deep shade) and, when present, outcompete small-seeded tree species which benefit from higher fecundity and colonization ability[70,71]. Larger trees better exploit below- and above-ground resources but can be more prone to drought-induced mortality[72] and stem uprooting and breakage by strong winds in forest edges[48], making them vulnerable to habitat fragmentation, wildfires, logging and defaunation[27]. Finally, the abundance of tree species whose seeds are dispersed by animal ingestion (that is, endozoochory) increase in disturbed Amazonian forests, while those species whose seeds are carried (but not ingested) by animals (that is, synzoochory) or dispersed by abiotic mechanisms (for example, wind) tend to decline[24], but the consistency of these patterns in other human-modified landscapes is yet to be tested.

The selected forest plots are those with at least 50% of species-level trait coverage (average = 85% across trait–region combinations) and

at least 80% of total trait coverage (average = 98%) of community abundance, the latter including genus-level trait data. The remaining missing values were imputed through multivariate trait imputation using the R package 'mice' version 3.16[73], but this represented <1% of the total number of individuals in all but 4 (<5%) of the 1,355 trait–plot combinations. Over one third of the studied species (438/1,207) have species-level data for all analysed traits. The distribution of species trait values covers similar ranges across the six study regions (Extended Data Fig. 8), despite variation in the multivariate functional space (that is, trait combinations) across regions (Extended Data Fig. 6a).

### Disturbance variables

Forest loss (that is, the percentage of the landscape covered by non-forest land uses), fragmentation (that is, number of forest patches) and edge density (that is, ratio of forest area less than 100 m away from the patch boundaries to the total forest area) were measured at three spatial scales, using concentric buffers of 500, 1,000 and 2,000 m radius from the centre of each sample plot (further referred to as 'local landscapes'; Fig. 2b)[33], in software QGIS version 3.22.14. The selected scales encompass and extend beyond those used in other studies of landscape-scale disturbance effects on tree assemblages (for example, ref. 25). We adopted this broad multiscale approach to assess the consistency of effects across scales as (1) there was no a priori information allowing us to select a single scale and (2) it was highly unlikely that a common 'scale of effect'[33] would emerge across regions, traits, predictors and response variables. Landscape metrics were based on 2010 data from the MapBiomas network (mapbiomas. org), whose classification is based on 30 m resolution Landsat images[74]. All study regions encompass local landscapes covering a large range of forest loss, number of patches and edge density (Extended Data Fig. 9). To assess additional effects of local degradation drivers, we used the inverse of tree community basal area (that is, the sum of cross-sectional area of all trees in a plot) relative to the sample area as a proxy. Basal area is one of the key determinants of above-ground biomass, which in turn is known to be strongly associated with different disturbances[34,35,65].

### Data analyses

We used a Structural Causal Modelling framework[30,31] to assess the total causal effect of each disturbance variable (forest loss, number of patches, edge density and local degradation) on the outcomes of interest, that is, community-level or species-level variables (see the following sections). The first step consisted in constructing a causal diagram (a DAG; Fig. 1), that is, a graphical representation of our causal ecological assumptions underlying the studied system. Causal thinking requires to account for both observed and unobserved but relevant variables or processes[75]. Here, a potential unobserved confounder of the represented cause–effect relations (Fig. 1) could be average climate and soil types. For example, at broad spatial scales going across regions and marked gradients, wetter climates could increase forest productivity and therefore forest degradation while independently controlling the functional assembly of tree communities (through spatial partitioning of species and life-history strategies relating to their fundamental and realized niches; for example, ref. 76). Such 'common cause' structures in a DAG can lead to spurious associations between the variable whose effect we investigate (for example, forest degradation) and the outcome (confounder bias)[31]. Here, to limit the risks of such biases, we performed the analysis of each causal effect within relatively homogeneous regions, climate- and soil-wise, to close potential non-causal paths of associations through such broad-scale unobserved variables. Our DAG does not represent this unobserved confounder structure and is therefore to be read as a 'within-region DAG'.

The second step consisted of investigating the DAG data consistency by testing the DAG's testable implications, that is, its conditional independencies[30]. However, our particular DAG did not have any testable implications. We next applied a logical graphical criterion—the

'backdoor criterion'—to the DAG, resulting for each variable whose causal effect we wanted to estimate (often referred to as 'exposure' variables) in the definition of a minimal set of control variables to allow a causal interpretation of the exposure's coefficient estimate. In other words, by conditioning a statistical model on the control variables defined by the backdoor criterion, this approach closes all non-causal paths between the exposure and outcome and only captures associations through causal paths (conditional on the DAG being correct).

Based on the DAG and the backdoor criterion, we therefore generated different models with different sets of control variables to estimate the total causal effects of each disturbance variable. As a result, the causal effects of the number of patches were estimated in models controlling for forest loss (that is, effect of 'fragmentation per se')[9], while the edge density models additionally account for fragmentation and local degradation models conditioned on all landscape-scale disturbance effects (the control variable sets of each model are presented in Fig. 3). Each of these models were further fitted separately for each trait in each region at each landscape scale. Before fitting our models, we log-transformed our metrics related to forest loss and number of patches, as well as seed mass, to reduce skewness in distributions.

### Community-level analysis

To describe community functional profiles, we measured abundance-weighted means (CWM) and 5% percentiles (lower and upper) of trait distributions, the first reflecting trait dominance and the latter extreme trait values in a community (Fig. 1, (1) in yellow shaded circle). Note that for all continuous functional traits (that is, excluding dispersal syndromes), low values reflect opportunistic strategies and high values more conservative traits (for example, we consider LMA instead of its inverse, specific leaf area, as higher values mean denser leaves). Similarly, all predictors considered are disturbance variables with values increasing with increasing level of disturbance, for example, forest loss instead of forest cover. Therefore, we expected negative relationships for all these trait–driver combinations (Fig. 1, yellow shaded circle).

After fitting our models following the causal framework described above, we identified and controlled for potential spatial autocorrelation in each model residuals, using Moran's Eigenvector Maps[77], a spatial eigenvector-based method allowing detecting simple to complex multiscale spatial patterns. We optimized the selection of a subset of spatial eigenvectors to be used as predictors together with the disturbance variables, based on the comparison of four spatial weighting matrices defined by the combination of two contrasted graph-based connectivity matrices (a Gabriel graph and a minimum spanning tree) and two weighting matrices (binary weighting or weighting decreasing linearly with distance between plots). The spatial weighting matrix comparison and selection were based on their statistical significance (corrected for multiple testing) and their respective smallest number of spatial eigenvectors necessary to capture all the residual spatial autocorrelation, following Bauman et al.'s recommendations[78,79]. This was done using the *adespatial* R package version 0.3-23[80].

### Species-level analysis

To assess the total causal effects of forest loss on species distribution, and the associated winner–loser trait replacements, we first assessed species-specific responses to forest loss in each region at each analysed scale. For this, we selected species occurring in at least five plots in each region, to allow a reliable estimate of their disturbance niche, resulting in 484 species and 765 species–region combinations, with species occurring on average in 14 to 85 plots within regions. For each of these species–region combinations, we calculated species' forest-loss niche centroid, defined as the average of forest loss across the plots where the species occurs, weighted by its abundance in each plot. We then applied a null model approach[36] in which we randomly distributed the abundance of species across plots within regions and re-calculated their niche centroid; we did so 10,000 times for each species–region combination. Standard

effect sizes (SES) were then calculated for each species in each region and at each local landscape scale, to describe the direction and magnitude of the deviation of observed niche centroid from random expectation, with positive and negative values reflecting species associated with more and less disturbed conditions, respectively. We considered species with observed niche centroid higher or lower than expected by chance (that is, higher or lower than 95% of the random values obtained from the 10,000 null-model iterations), as potential winners or losers, respectively. This method has been successfully applied to identify winners and losers of land-use intensification[36] and has the advantage of being based on abundance-weighted niche centroids, allowing species that have reduced abundance in deforested landscapes to be defined as losers and species that are usually present but thrive in disturbed sites as winners.

Finally, we applied linear models to test the effects of species trait values on their SES forest-loss niche centroid, using a similar causal modelling framework as for the community-level analyses (above). In addition, we illustrate trait differences among winner and loser species and test them by means of analysis of variance. Better model performance or more pronounced winner–loser trait differences when including weights for species total abundance across plots was taken to be indicative of community-level changes being driven by the turnover of a few dominant species. It is worth noting that the species-level analyses were focused on the total causal effect of forest loss, as it had the strongest effect at the community level and the causal effects of other disturbance variables were rare or comparatively smaller at the species level. All the analyses described above were performed in software R 4.3.2[81].

## Multivariate analysis across biological levels

To complement, summarize and further test the robustness of these analyses, we applied a dc-CA to all data (vegetation, traits and landscape variables at three spatial scales in all six regions) simultaneously, in a way that is very similar in spirit to the previous community- and species-level analyses[37]. The dc-CA method is a constrained ordination method which extends canonical correspondence analysis in that it constrains not only the site scores by predictors but also the species scores by traits[37]. By combining CWM regression at the community level and species niche centroids-trait regression at the species level, dc-CA finds the best composite trait. This composite trait represents the linear combination of traits, whose CWMs are best explained by linear regression onto the predictors, the fitted values of which form the dc-CA axis[37]. To guard against the type 1 error rate inflation that is typical for CWM regression[28,54], dc-CA performs statistical tests by taking the maximum of the P values of the community-level and species-level permutation tests, that is, by the max test[54,82]. In summary, dc-CA applies dimension reduction to a multitrait CWM regression with a guard against over-optimism in statistical tests. For more details, see refs. 49,83.

Initially, all three landscape variables at each analysed spatial scale and local degradation were each used as a single 'environmental' predictor in dc-CA using the software Canoco version 5.15[84]. Subsequently, dc-CA was used with the same sequence of analyses as in the DAG-based community-level analysis (Fig. 3) and with selection of scale of each landscape variable that best explained the within-region differences in CWMs of all seven traits[83]. A single dc-CA analysis summarized the main results. We divided the abundance data by the plot totals before analysis, so that the CWM regressions in dc-CA are also unweighted, with the additional advantage of reducing putative plot-size effects. In the selection of best scale, the Bonferroni procedure was used to correct for multiple statistical testing. The dc-CA was forced to analyse the within-region differences—following the reasoning underlying the causal structural modelling analytical framework—by setting the factor 'region' as a covariate (Condition in the formula version of cca in the vegan R package) and by testing the same sequence of models that are shown in Fig. 3. Regional dependence of relationships was investigated by adding interactions of the main predictor(s) with region. For reproducibility, an R library for dc-CA was created[85].

## Reporting summary

Further information on research design is available in the Nature Portfolio Reporting Summary linked to this article.

## Data availability

All data used in the analysis are available via figshare at https://doi.org/10.6084/m9.figshare.25565169 (ref. 86). These data result from the work of several people who applied for grants, sampled the tree plots and kept long-term plots running at great expenses. As such, it would be appreciated if data owners were consulted and invited for any publications using this dataset.

## Code availability

All code used for analysis is available via figshare at https://doi.org/10.6084/m9.figshare.25565169 (ref. 86).

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

## Acknowledgements

The writing and analysis in this study were supported by Natural Environment Research Council Arboles project NE/S011811/1. Research in Amazonia was funded by Instituto Nacional de Ciência e Tecnologia—Biodiversidade e Uso da Terra na Amazônia (Conselho Nacional de Desenvolvimento Científico e Tecnológico - CNPq 574008/2008-0), Programa de Pesquisas Ecológicas de Longa Duração Rede Amazônia Sustentável - PELD-RAS (CNPq 441573/2020-7), Empresa Brasileira de Pesquisa Agropecuária—Embrapa (SEG:02.08.06.005.00), the UK government Darwin Initiative (number 17-023), The Nature Conservancy (NE/F01614X/1 and NE/G000816/1), the Rufford Small Grant Foundation (9856-1), the Conservation Food and Health Foundation and Idea Wild. B.X.P. acknowledges funding from CNPq (number 140260/2015-3) and B.X.P. and I.M. from the French National Research Agency (ANR) under the 'Investissements d'avenir' program (ANR-16-IDEX-0006 and ANR-10-LABX-25-01). M.B received a productivity grant from CNPq (number 304189/2022-7). D.B. was funded by the European Union's Horizon 2020 research and innovation program under the Marie Skłodowska-Curie grant agreement (number 895799). We thank D. Jamelli for measuring landscape variables, N. Blüthgen for insights on the null models for detecting winner and loser species and J. Oliveira for drawing the conceptual diagram in Fig. 1.

## Author contributions

B.X.P., F.P.L.M. and J.B. conceived the ideas and designed methodology, with the support of D.B. in the definition of the causal inference analytical framework. B.X.P. compiled and analysed the data, with the support of C.J.F.t.B. on the dc-CA analysis. B.X.P. wrote the first draft, with substantial contributions from J.B. J.B, F.P.L.M., I.M. and M.T. acquired funds and supervised the work. All authors contributed data and critical insights that improved the manuscript.

## Competing interests

The authors declare no competing interests.

## Additional information

**Extended data** is available for this paper at https://doi.org/10.1038/s41559-024-02592-5.

**Correspondence and requests for materials** should be addressed to Bruno X. Pinho or Jos Barlow.

[1]Institute of Plant Sciences, University of Bern, Bern, Switzerland. [2]AMAP, Univ Montpellier, CIRAD, CNRS, INRAE, IRD, Montpellier, France. [3]Departamento de Botânica, Universidade Federal de Pernambuco, Recife, Brazil. [4]School of Animal, Rural and Environmental Sciences, Nottingham Trent University, Nottingham, UK. [5]Biometris, Wageningen University & Research, Wageningen, The Netherlands. [6]Laboratoire d'Écologie Végétale et Biogéochimie, Université Libre de Bruxelles, Brussels, Belgium. [7]Environmental Change Institute, University of Oxford, Oxford, UK. [8]Applied Ecology and Conservation Lab, Universidade Estadual de Santa Cruz, Ilhéus, Brazil. [9]Escuela Nacional de Estudios Superiores, Universidad Nacional Autónoma de México, Mérida, Mexico. [10]Instituto de Investigaciones en Ecosistemas y Sustentabilidad, Universidad Nacional Autónoma de México, Morelia, Mexico. [11]Departamento de Sistemática e Ecologia, Universidade Federal da Paraíba, João Pessoa, Brazil. [12]Institute of Science and Environment, University of Cumbria, Ambleside, UK. [13]Instituto Juruá, Manaus, Brazil. [14]Brazilian Agricultural Research Corporation (EMBRAPA), Belém, Brazil. [15]Lancaster Environment Centre, Lancaster University, Lancaster, UK. [16]School of Environmental Sciences, University of East Anglia, Norwich, UK. [17]Departamento de Ecologia e Conservação, Instituto de Ciências Naturais, Universidade Federal de Lavras, Lavras, Brazil. [18]Institute for Global Change Biology, School for Environment and Sustainability, University of Michigan, Ann Arbor, MI, USA. [19]Department of Ecology and Evolutionary Biology, University of Michigan, Ann Arbor, MI, USA. [20]Instituto de Biologia, Universidade Federal da Bahia, Salvador, Brazil. ✉e-mail: bxpinho@hotmail.com; josbarlow@gmail.com

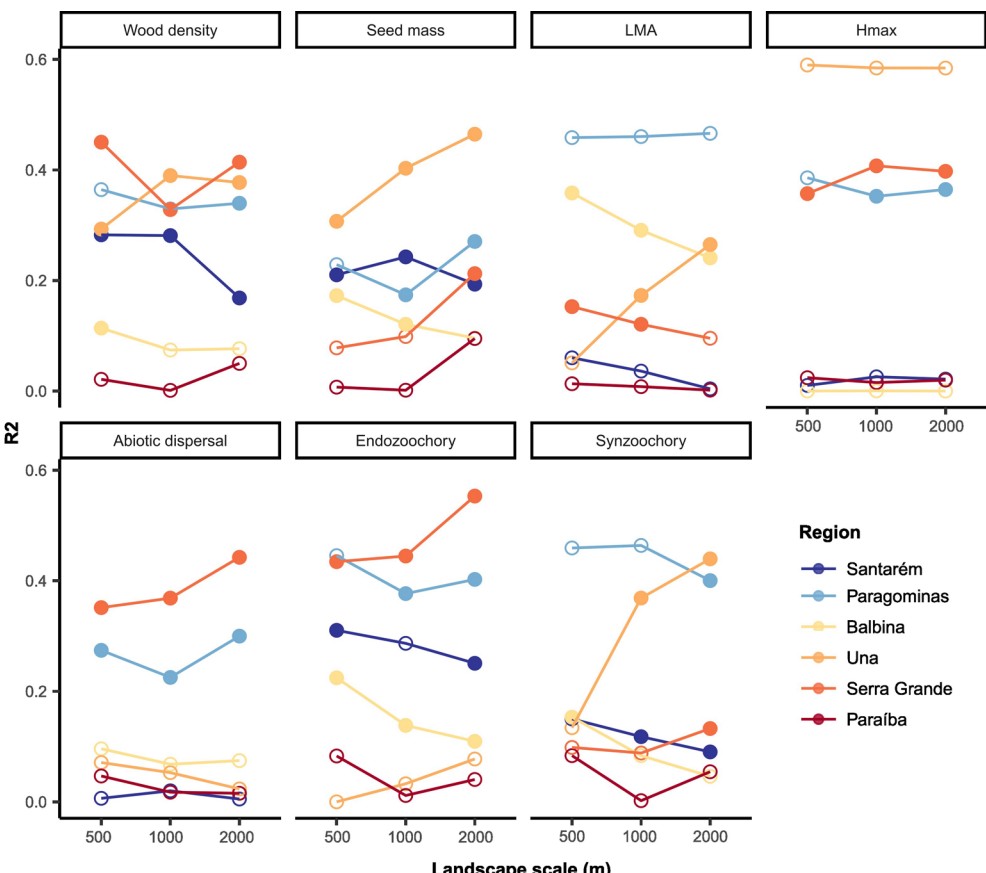

**Extended Data Fig. 1 | Scale of forest-loss effects on CWM traits.** Explained variation (R2) in models predicting community-weighted trait means (CWMs) from forest loss at each analyzed landscape scale (x-axis) for each analyzed trait (panels) in each study region (colors). Filled points denote significant relationships. LMA = leaf mass per area.

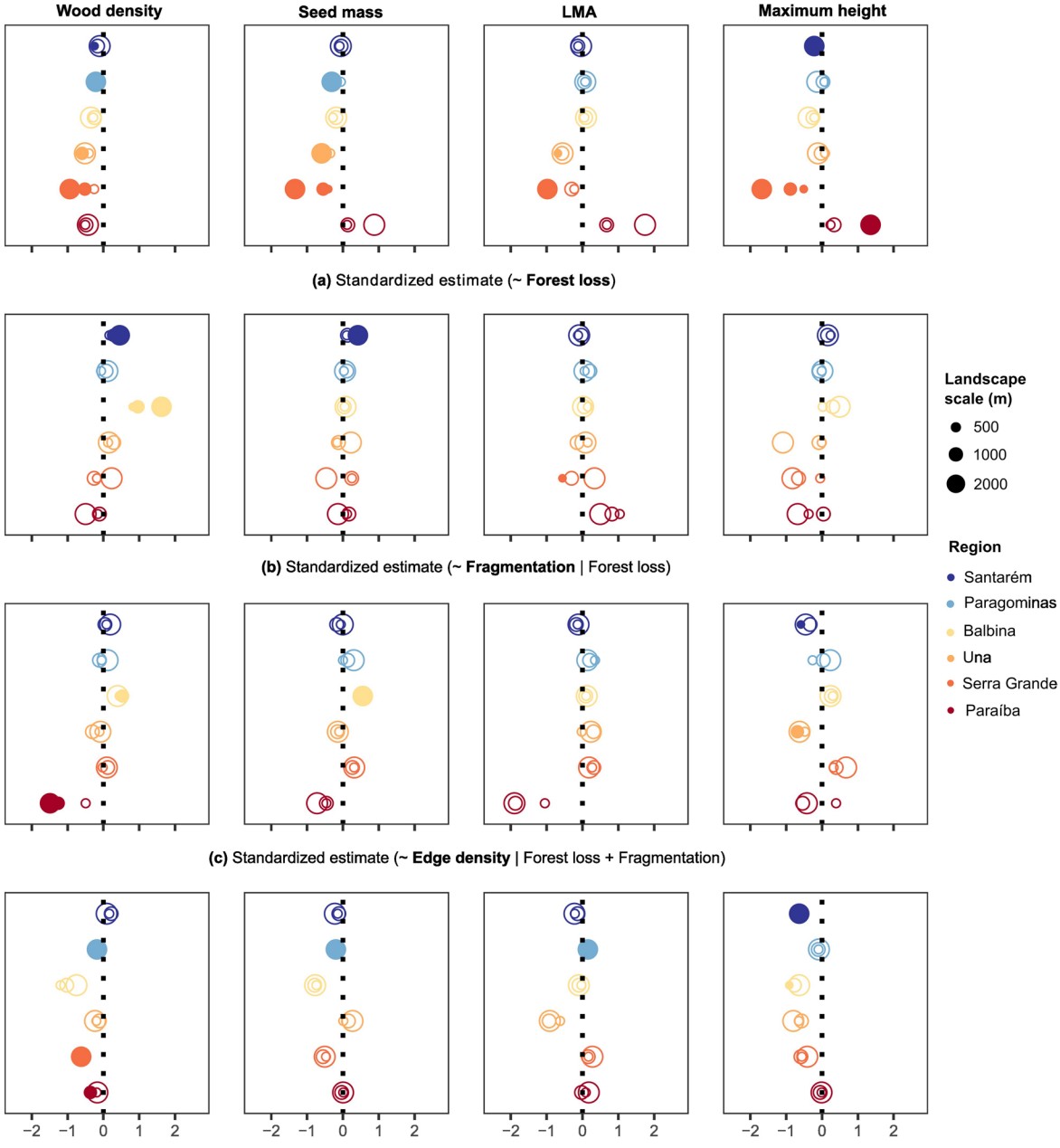

**Extended Data Fig. 2 | Effects of landscape and local drivers on upper 5% percentiles of community trait distributions.** Direction and magnitude of the causal effects of (a) landscape-scale forest loss, (b) fragmentation (that is, number of patches), (c) edge density and (d) local degradation (that is, the inverse of tree community basal area) on upper 5% percentiles of tree community trait distributions across 271 old-growth forest plots in six neotropical forest regions (colors) in the Amazonian and Atlantic forests, Brazil (Fig. 1). Landscape drivers were measured at multiple spatial extents (circle sizes) surrounding each sample plot (Fig. 2b). Predictors and traits were standardized, so that comparing causal effect estimates among the four models reflects *relative* trait responses to increases of one standard deviation in the predictor of interest (in bold). Filled points denote significant effects (that is, those with estimates 95% confidence intervals non-overlapping the zero effect). Regions are ordered from the least to the most disturbed (top-down) according to overall forest cover and land-use history (Extended Data Table 1). LMA, leaf mass per area.

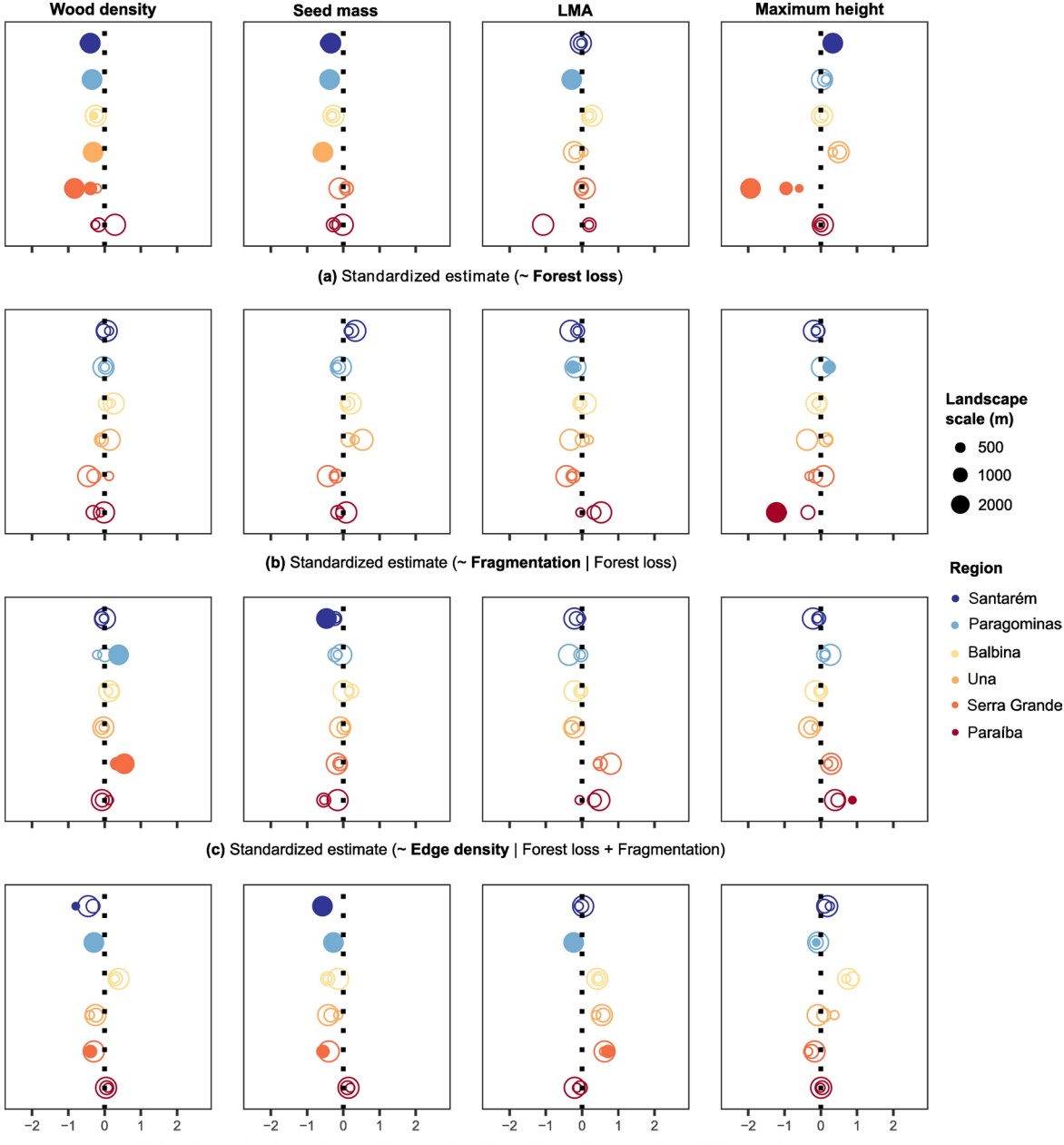

**Extended Data Fig. 3 | Effects of landscape and local drivers on lower 5% percentiles of community trait distributions.** Direction and magnitude of the causal effects of (a) landscape-scale forest loss, (b) fragmentation (that is, number of patches), (c) edge density and (d) local degradation (that is, the inverse of tree community basal area) on lower 5% percentiles of tree community trait distributions across 271 old-growth forest plots in six neotropical forest regions (colors) in the Amazonian and Atlantic forests, Brazil (Fig. 1). Landscape drivers were measured at multiple spatial extents (circle sizes) surrounding each sample plot (Fig. 2b). Predictors and traits were standardized, so that comparing causal effect estimates among the four models reflects *relative* trait responses to increases of one standard deviation in the predictor of interest (in bold). Filled points denote significant effects (that is, those with estimates 95% confidence intervals non-overlapping the zero effect). Regions are ordered from the least to the most disturbed (top-down) according to overall forest cover and land-use history (Extended Data Table 1). LMA, leaf mass per area.

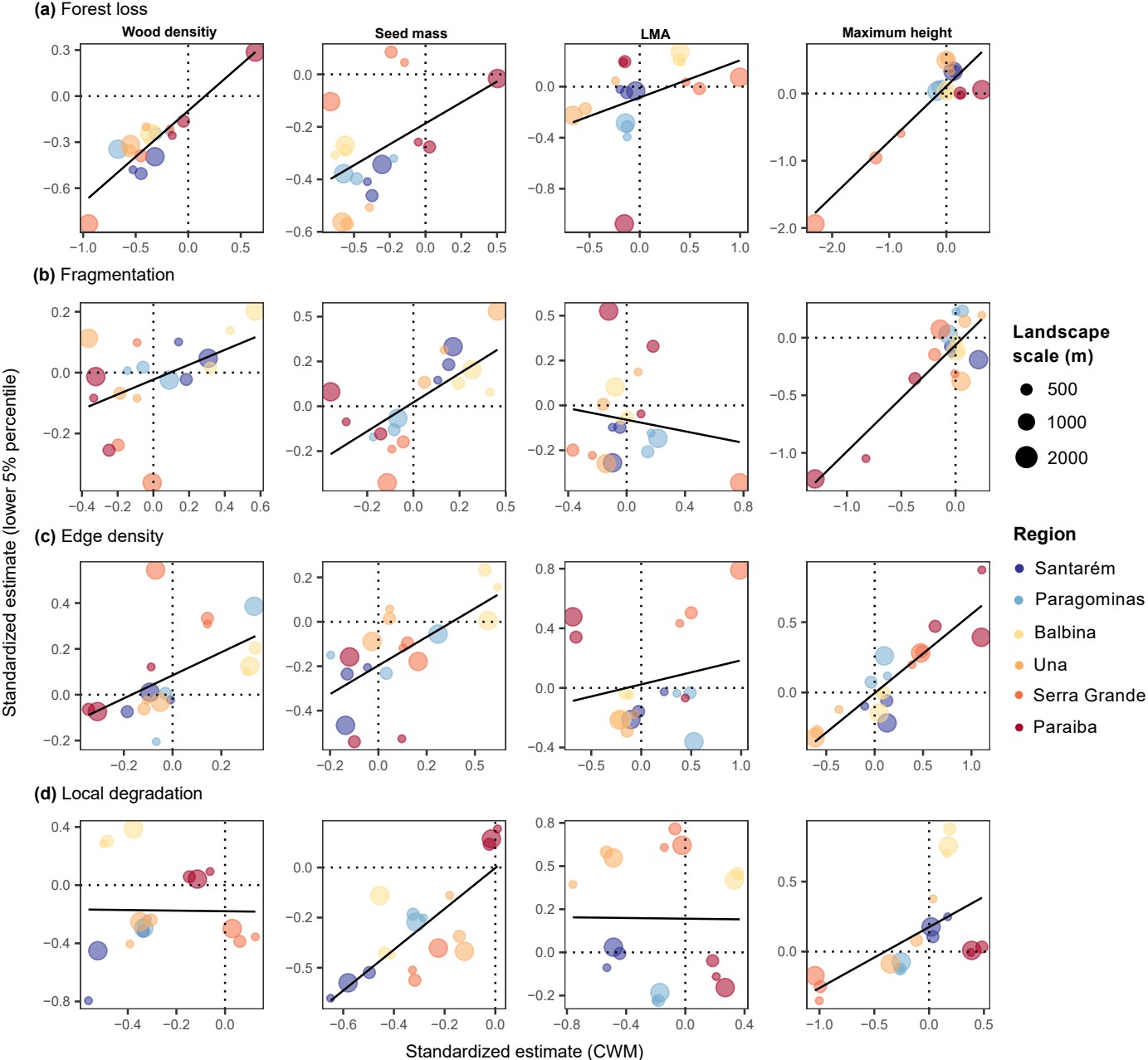

**Extended Data Fig. 4 | Correlations between observed effects on CWMs and lower 5% percentiles of community trait distributions.** Correlations between standardized causal effect estimates (that is, effect sizes) of models predicting abundance-weighted means (x-axis) and lower 5% percentiles (y-axis) of community trait distributions in response to (a) forest loss, (b) fragmentation, (c) edge density, and (d) local degradation, for each study region (point colors) at each analyzed landscape scale (point sizes). Traits are organized by columns. Models for the total causal effect of each predictor have specific sets of control variables (see in Fig. 3). Points in the bottom-left and upper-right sides of each panel (as separated by dashed lines showing zero effects) illustrate similar effects in direction, while those in the bottom-right and upper-left sides illustrate contrasting effects. LMA, leaf mass per area.

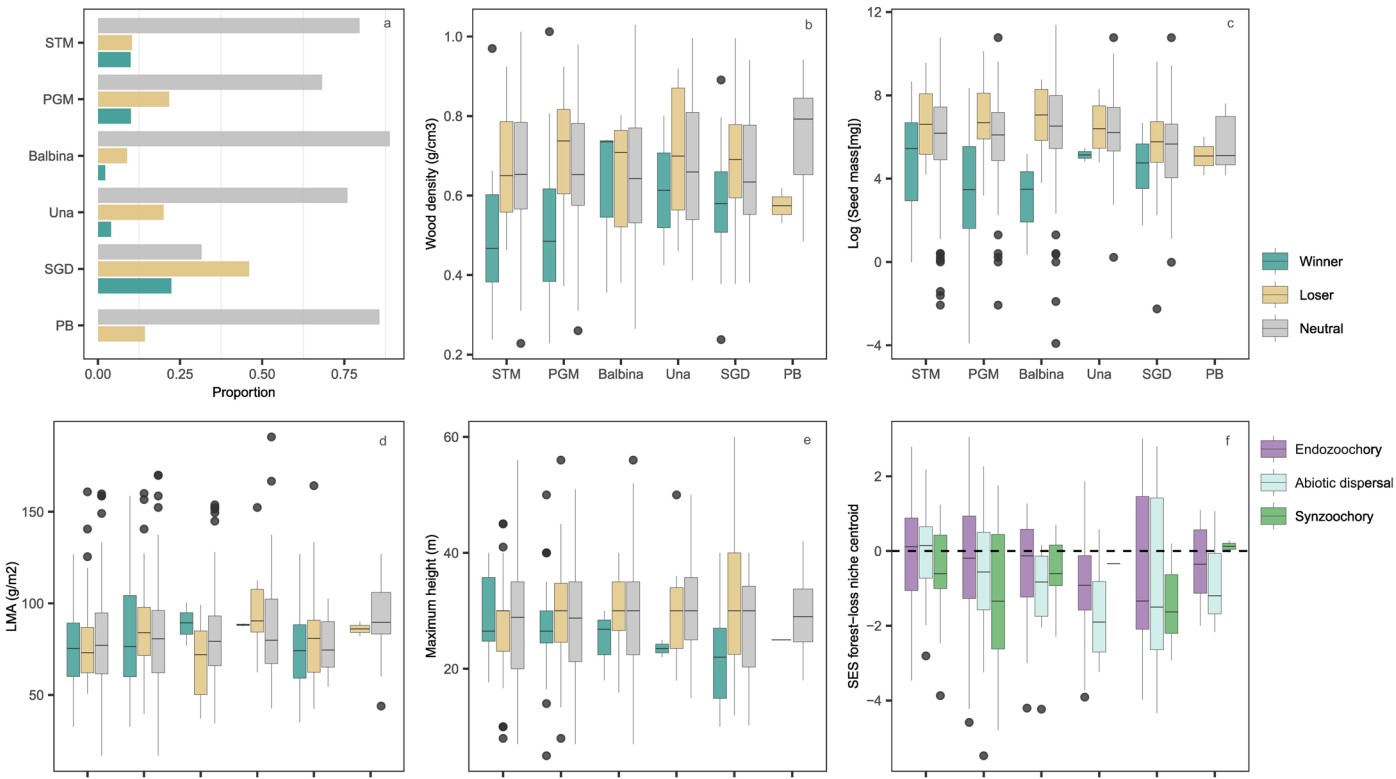

**Extended Data Fig. 5 | Proportion and trait distributions of winner and loser species.** Proportion of species that are winners, losers or have neutral distribution along forest loss gradients in each of the six study regions (**a**), and trait differences among them (**b**-**f**). Note that for the categorical trait, dispersal syndrome (**f**), we illustrate it by showing in the y-axis the deviation of species niche centroids from random expectation, that is the standard effect size (SES), where positive and negative values represent species significantly associated with more or less deforested landscapes, respectively. Regions are ordered from the least to the most disturbed (left-right) according to overall forest cover and land-use history (Extended Data Table 1). STM, Santarém; PGM, Paragominas; SGD, Serra Grande; PB, Paraíba.

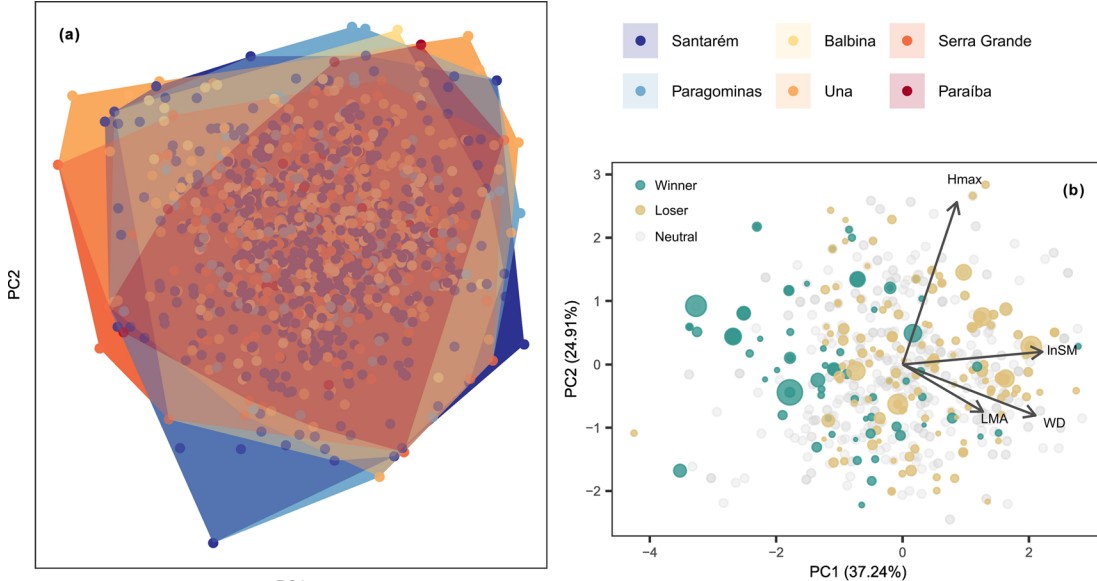

**Extended Data Fig. 6 | Principal component analysis of tree species functional traits across the six study regions.** Ordination diagram of the first two axes of the principal component analysis (PCA) of tree species trait values, for (**a**) all species (N = 1,207 species and 2,520 species-region combinations) in six neotropical forest regions (colored polygons), and (**b**) only species occurring in at least five plots in a given region (N = 484 species and 765 species-region combinations) which were included in our niche analysis, with colors representing their fate with forest loss: winners (green points) are those species whose distribution (that is abundance-weighted niche optimum) along forest loss gradients significantly deviate positively from random expectation, while losers (yellow points) are those whose abundance significantly decreases in communities embedded in more deforested landscapes, and neutral species (gray points) do not show significant response to forest loss. In (**b**), points representing winners and losers increase in size with species total abundance across plots, to highlight trait differences between *dominant* winner-loser species. Arrows indicate the direction and weighting of vectors representing the four continuous traits considered: WD = wood density, lnSM = logarithmic of seed mass, LMA = leaf mass per area, Hmax = maximum height.

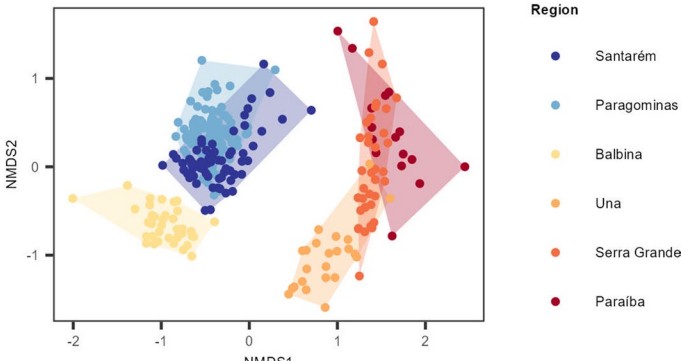

**Extended Data Fig. 7 | Variation in tree species composition across study plots and regions.** Non-metric multidimensional scaling (NMDS) ordination of tree taxonomic compositional variation (Chao–Jaccard dissimilarities) among 271 old-growth forest plots in six neotropical forest regions (colors) in the Brazilian Amazonian and Atlantic forests (see Fig. 1).

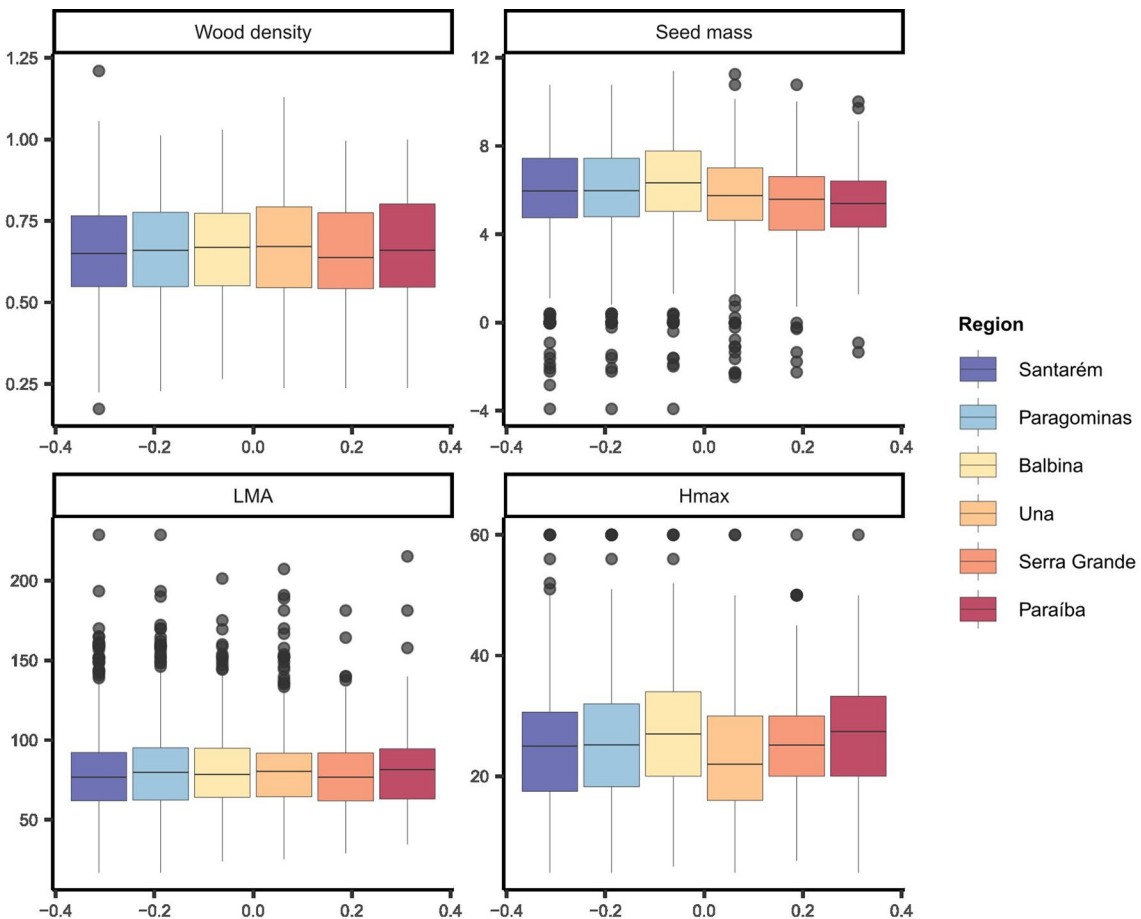

**Extended Data Fig. 8 | Variation in tree species functional traits in each study regions.** Distribution of tree species trait values (n = 1,207 species) in the six study regions. Boxplots indicate the median (center line), 25–75% quartiles (box edges), <1.5 times the interquartile range (whiskers), and extreme values (dots). Regions are ordered from the least to the most disturbed (top-down in legend) according to overall forest cover and land-use history (Extended Data Table 1). LMA = leaf mass per area.

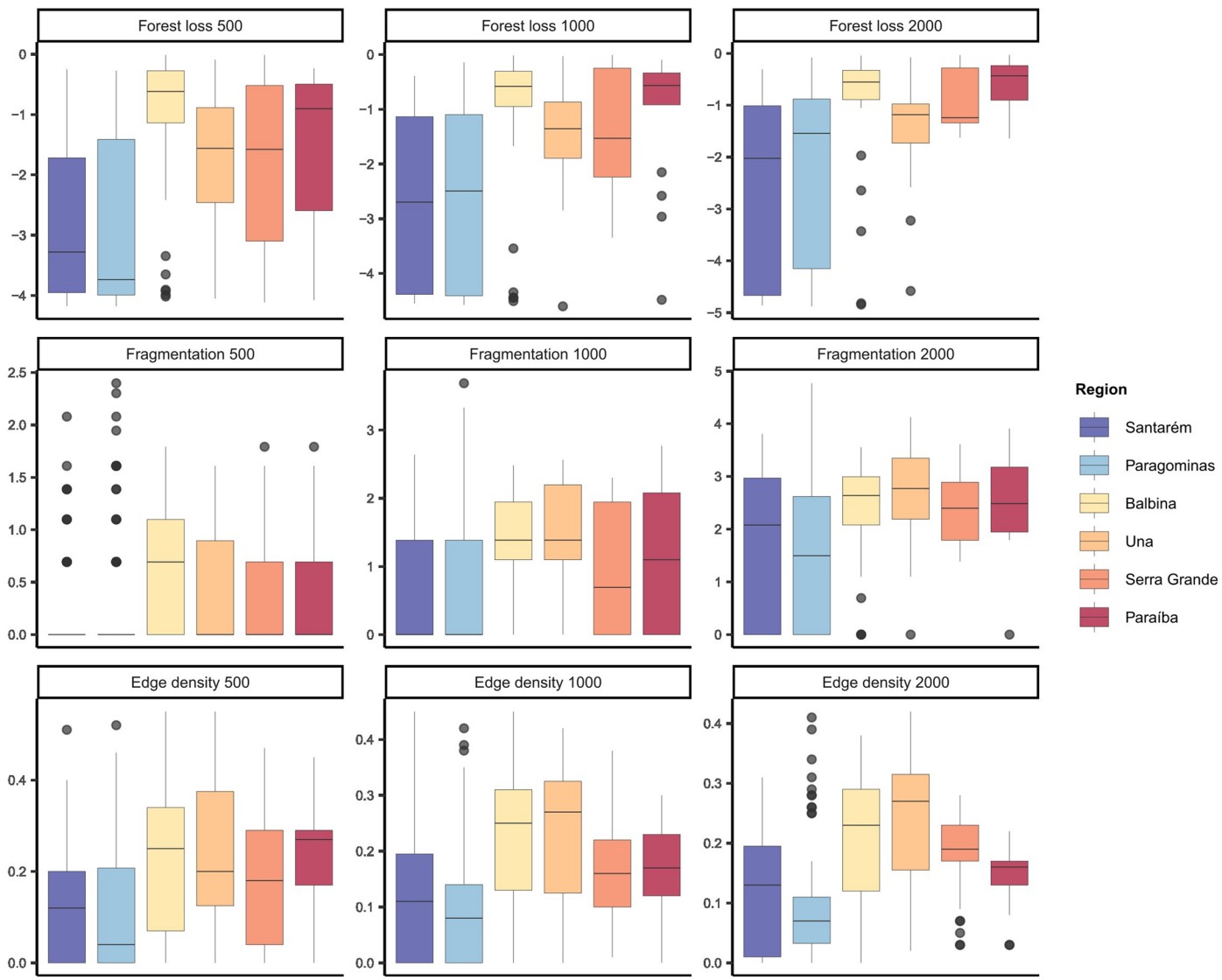

**Extended Data Fig. 9 | Variation in landscape-scale predictors across scales and regions.** Distribution of landscape-scale predictors (rows) at each analyzed scale (columns) in the six study regions (colors). Boxplots indicate the median (center line), 25–75% quartiles (box edges), < 1.5 times the interquartile range (whiskers), and extreme values (dots). Regions are ordered from the least to the most disturbed (top-down) according to overall forest cover and land-use history (Extended Data Table 1).

**Extended Data Table 1 | Description of the study regions**

| Region | Location (state) | N plots | Plot size (ha) | MAP (mm) | Time disturbance (~ yrs) | Deforestation level | Defaunation level | Dominant matrix | Refs* |
|---|---|---|---|---|---|---|---|---|---|
| Santarém | Pará | 63 | 0.25 | 1875 | 40 | Intermediate | Low | Heterogeneous | 1 |
| Paragominas | Pará | 94 | 0.25 | 1962 | 40 | Intermediate | Low | Heterogeneous | 1,2 |
| Balbina | Amazonas | 37 | 0.25 | 2361 | 40 | Intermediate | Low | Water | 3 |
| Una | Bahia | 23 | 0.05–0.08 | 1602 | 50 | High | Intermediate | Cocoa | 4,5 |
| Serra Grande | Alagoas | 37 | 0.1 | 1610 | 300 | High | High | Sugar cane | 6,7 |
| Paraiba | Paraiba | 23 | 0.03–0.10 | 1611 | 300 | Very high | Very high | Urban | 8,9 |

Location, sampling effort, average precipitation and land-use history and patterns of the six study regions distributed across the Amazonian and Atlantic forests in Brazil. MAP = mean annual precipitation. *References: 1. Berenguer et al.[34], 2. Berenguer et al.[22], 3. Benchimol & Peres[60], 4. Pardini et al. (2009), 5. Faria et al.[66], 6. Santos et al.[13], 7. Mendes et al. (2016), 8. Pinho et al.[65], 9. Junior et al. (2017).

# Reporting Summary

## Statistics

For all statistical analyses, confirm that the following items are present in the figure legend, table legend, main text, or Methods section.

| n/a | Confirmed | |
|---|---|---|
| ☐ | ☒ | The exact sample size (*n*) for each experimental group/condition, given as a discrete number and unit of measurement |
| ☐ | ☒ | A statement on whether measurements were taken from distinct samples or whether the same sample was measured repeatedly |
| ☐ | ☒ | The statistical test(s) used AND whether they are one- or two-sided *Only common tests should be described solely by name; describe more complex techniques in the Methods section.* |
| ☐ | ☒ | A description of all covariates tested |
| ☐ | ☒ | A description of any assumptions or corrections, such as tests of normality and adjustment for multiple comparisons |
| ☐ | ☒ | A full description of the statistical parameters including central tendency (e.g. means) or other basic estimates (e.g. regression coefficient) AND variation (e.g. standard deviation) or associated estimates of uncertainty (e.g. confidence intervals) |
| ☐ | ☒ | For null hypothesis testing, the test statistic (e.g. *F*, *t*, *r*) with confidence intervals, effect sizes, degrees of freedom and *P* value noted *Give P values as exact values whenever suitable.* |
| ☒ | ☐ | For Bayesian analysis, information on the choice of priors and Markov chain Monte Carlo settings |
| ☐ | ☒ | For hierarchical and complex designs, identification of the appropriate level for tests and full reporting of outcomes |
| ☐ | ☒ | Estimates of effect sizes (e.g. Cohen's *d*, Pearson's *r*), indicating how they were calculated |

*Our web collection on statistics for biologists contains articles on many of the points above.*

## Software and code

Policy information about availability of computer code

| | |
|---|---|
| Data collection | All the data used in this study were either collected by the authors in previous work or obtained from various publicly available sources as described in the Methods section. Restructuring of the data prior to the analyses was done in the R coding environment. |
| Data analysis | All the data analyses were performed in software R version 4.3.2. All the R code necessary to reproduce the results and figures of the present study were shared for review and are available via figshare at https://doi.org/10.6084/m9.figshare.25565169 |

For manuscripts utilizing custom algorithms or software that are central to the research but not yet described in published literature, software must be made available to editors and reviewers. We strongly encourage code deposition in a community repository (e.g. GitHub). See the Nature Portfolio guidelines for submitting code & software for further information.

## Data

Policy information about availability of data

All manuscripts must include a data availability statement. This statement should provide the following information, where applicable:
- Accession codes, unique identifiers, or web links for publicly available datasets
- A description of any restrictions on data availability
- For clinical datasets or third party data, please ensure that the statement adheres to our policy

All data used in the analysis are available via figshare atAll data and code required to reproduce the results of this study were shared for review and will be available

# Research involving human participants, their data, or biological material

Policy information about studies with human participants or human data. See also policy information about sex, gender (identity/presentation), and sexual orientation and race, ethnicity and racism.

| | |
|---|---|
| Reporting on sex and gender | This study does not involve human-related data, therefore no gender analysis were performed. |
| Reporting on race, ethnicity, or other socially relevant groupings | This study does not address human-related issues, therefore no social groupings were used. |
| Population characteristics | This study does not include any human-related data, therefore no population characteristics were described. |
| Recruitment | This study did not involve human participants, therefore no recruitment was done. |
| Ethics oversight | No specific ethic protocol was used as the present study does not involve any ethical issues and does not include any human-related data. |

Note that full information on the approval of the study protocol must also be provided in the manuscript.

# Field-specific reporting

Please select the one below that is the best fit for your research. If you are not sure, read the appropriate sections before making your selection.

☐ Life sciences          ☐ Behavioural & social sciences          ☒ Ecological, evolutionary & environmental sciences

For a reference copy of the document with all sections, see nature.com/documents/nr-reporting-summary-flat.pdf

# Ecological, evolutionary & environmental sciences study design

All studies must disclose on these points even when the disclosure is negative.

| | |
|---|---|
| Study description | This study applies a causal inference framework using extensive floristic and plant trait data to investigate how consistent and predictable are the causal effects of landscape-scale forest loss, landscape configuration and local degradation on the functional profiles of tree assemblages in human-modified tropical forests. |
| Research sample | We used abundance and trait data from 1,207 tree species across 271 tropical forest plots in six human-modified regions of the Amazonian and Atlantic forests in Brazil. |
| Sampling strategy | The present study did not involve sampling of novel observations. All the data were either previous collected by the authors or retrieved from online data repositories as described in the methods section. |
| Data collection | All the floristic data were previously collected by the authors following standard vegetation inventory designs, including the identification of all trees with diameter at breast height (DBH) > 10 cm in forest plots of varying sizes (see Extended Data Table 1). The trait data were either collected by the authors or retrieved from online data repositories, mainly the TRY database. The data on land-use types used to calculate landscape metrics were retrieved from the MapBiomas public repository. |
| Timing and spatial scale | This study includes data from six Neotropical forest regions, which were collected at different times, between 2000 and 2015. However, we did not perform any temporal analyses, as we relied on one single temporal snapshot in each region. The spatial scale of the analyses was within regions, which comprised 18 to 87 forest plots. |
| Data exclusions | Exclusion criteria was adopted based on the trait data coverage of total plot-level tree abundance. The selected forest plots are those with at least 50% of species-level trait data coverage and at least 80% of total trait coverage of community abundance, the latter including genus-level trait data. |
| Reproducibility | This study did not include an a priori experimental design, but all the steps of our data analyses and modeling approach are detailed in the Methods section and reproducible based on the data and code shared in a public repository. |
| Randomization | This study did not include an a priori experimental design, and therefore randomization was not necessary. |
| Blinding | This study did not include an a priori experimental design, and therefore blinding was not necessary. |

Did the study involve field work?          ☐ Yes          ☒ No

# Reporting for specific materials, systems and methods

We require information from authors about some types of materials, experimental systems and methods used in many studies. Here, indicate whether each material, system or method listed is relevant to your study. If you are not sure if a list item applies to your research, read the appropriate section before selecting a response.

## Materials & experimental systems

| n/a | Involved in the study |
|-----|----------------------|
| ☒ ☐ | Antibodies |
| ☒ ☐ | Eukaryotic cell lines |
| ☒ ☐ | Palaeontology and archaeology |
| ☒ ☐ | Animals and other organisms |
| ☒ ☐ | Clinical data |
| ☒ ☐ | Dual use research of concern |
| ☒ ☐ | Plants |

## Methods

| n/a | Involved in the study |
|-----|----------------------|
| ☒ ☐ | ChIP-seq |
| ☒ ☐ | Flow cytometry |
| ☒ ☐ | MRI-based neuroimaging |

## Plants

| | |
|--|--|
| Seed stocks | This study did not use seed stocks or involved the collection of any plant materials, only previously sampled data were used. |
| Novel plant genotypes | This study did not produce or used novel plant genotypes. |
| Authentication | No seed stocks or novel genotypes were generated or used in this study, therefore no authentication was needed. |

