## [Peer Review File · Nature Ecology & Evolution]

Winner-loser plant trait replacements in human-modified tropical forests

Corresponding Author: Dr Bruno Pinho

Version 0:

Decision Letter:

Dear Dr Pinho,

Thank you for your patience while your manuscript entitled "Winner-loser plant trait replacements in human-modified tropical forests" was under review. We have now received three reports, appended at the end of this email. As you will see, the reviewers find the work of interest but raise a number of concerns that will need to be addressed before we can offer publication in Nature Ecology & Evolution. We therefore invite you to revise your manuscript taking into account all reviewer and editor comments.

Please note that I have contacted Reviewer 3 to ask for a clarification on their expectations from the revisions, as they are not very clear to us from the report. As we have received no reply and, based on the collective advice received, it seemed unlikely that we would reject the manuscript without inviting a revision at this stage, my colleagues and I decided to proceed without further delays. I will forward you Reviewer 3's answer if we receive it.

* Please highlight all changes in the manuscript text file.

* If you have not done so already please begin to revise your manuscript so that it conforms to our Article format instructions at <http://www.nature.com/natecolevol/info/final-submission>. Refer also to any guidelines provided in this letter.

Link Redacted

We hope to receive your revised manuscript within the next two months. If you need substantially more time, please let us know. We will be happy to consider your revision so long as nothing similar has been accepted for publication at Nature Ecology & Evolution or published elsewhere.

Nature Ecology & Evolution is committed to improving transparency in authorship. As part of our efforts in this direction, we are now requesting that all authors identified as 'corresponding author' on published papers create and link their Open Researcher and Contributor Identifier (ORCID) with their account on the Manuscript Tracking System (MTS), prior to acceptance. ORCID helps the scientific community achieve unambiguous attribution of all scholarly contributions. You can create and link your ORCID from the home page of the MTS by clicking on 'Modify my Springer Nature account'. For more information please visit www.springernature.com/orcid.

[redacted]

Reviewers' comments:

Reviewer #1 (Remarks to the Author):

This study examines the winner-loser trait spectrum under forest loss the Amazon and Atlantic forests and seeks to determine which trait changes are associated with several metrics/dimensions of anthropogenic forest modification. The study's major strength is that it presents a data synthesis large enough to explore whether winner-loser trait changes are consistent across this broad region. The analysis is detailed and interpreted carefully. The results show quite consistent impacts of forest loss on a subset of the traits measured, but no particularly strong and consistent causal effects of three descriptors of forest landscape change. Apart from a few specific comments, my one larger comment is that the authors could make more of the detailed results to help disentangle which ecological drivers (rather than landscape metrics) cause the winner-loser spectrum.

The anthropogenic drivers that the study focuses on (fragmentation, edge density, local degradation) are not really ecological processes that directly alter plant functional trait distributions. The paper does mention several ecological processes that have direct demographic effects on plants: biophysical edge effects (the authors discuss wind/storm damage, but heat/desiccation are relevant too), fire, and alteration of seed dispersal. Although I don't expect this study to incorporate data on these processes, I think the Discussion section could better leverage the results to disentangle which of these ecological drivers are stronger / weaker mechanistic explanations for the observed patterns. This could help orient future research on these direct ecological drivers causing this winner-loser spectrum.

For example, if the prediction at L422 that "trees with high densities can be especially vulnerable to stem breakage with extreme winds in edge-affected habitats" were true, then we should expect negative values for wood density primarily associated with Edge Density (Fig. 3C) and Fragmentation (Fig. 3B). Because the only significant values are instead positive, this ecological mechanism does not appear to be supported. Similarly, if leaf economics traits mediated the changes in edge/disturbed habitats, then we would probably expect traits like LMA to respond consistently to edge density and local degradation in particular, which also wasn't observed. If the major driver were fire, which often affects forests near edges, then we might expect the largest significant effects across traits to be associated with edge density rather than just forest loss – again something not supported by these results. In contrast to some of the other drivers that should be most strongly associated with edge density etc, I expect loss of large-bodied seed dispersers to be primarily determined by habitat loss (forest loss) and to a lesser extent fragmentation, and thus expect their effects to be borne out in panel A, which is consistent with the results. Also because dispersal traits and traits linked to dispersal by large-bodied seed dispersers (large-seeded, high density woods) are the traits most consistently associated with the winner-loser spectrum, it appears that loss of large dispersers in human-modified tropical forests is the best supported ecological driver of the winner-loser spectrum.

The authors would likely make other/different interpretations, but including in the Discussion and perhaps in the Abstract some reflections along these lines would help this paper make a larger advance, including by potentially helping move this field past the focus on disentangling the effects of different measures of landscape modification (number of patches vs edge density), which are analyzed as potential proxies of ecological drivers, and toward a better understanding of which direct ecological drivers are most important and of their functional effects in human-modified forest landscapes.

Specific comments

L224. At the risk of adding further figures, I'm unsure why only the upper 95% percentile was considered and not a lower percentile as well in this case. For example, an increasing prevalence of very small-seeded / weedy species is potentially equally meaningful as change in the largest seed sizes, and while that's reflected in some of the other analysis, it isn't shown here.

Figure 3. It would help readers to provide a description of the (~ A | B) notation within the figure legend. Or in the text walk readers through the relationship to the causal links drawn in Fig 1 (ie in order to assess local degradation, the analysis needs to account for the effects of forest loss, fragmentation, and edge density). Figure 4's caption provides more explanation of this than does Figure 3.

There are some cited papers missing from the References list.

Reviewer #2 (Remarks to the Author):

This study explored the independent effects of forest loss, fragmentation, and degradation on plant functional composition in tropical rainforests with distinct histories and intensities of land use. They conclude that forest loss has a dominant role in driving functional trait composition, and local degradation can amplify the effects of forest loss. However, against most case studies that did not distinguish the effect of fragmentation per se, they found that the effect of landscape configuration on functional trait composition is rare.

This interesting study has a large dataset involving forest landscapes (area, number of fragments, and edge density), species composition, community structure (basal area) and functional traits from 271 forest plots in six regions of Amazon. Therefore, the results from this study can make great contributions to test whether previous case studies on this topic (e.g., Magnago et al. 2014) is context-dependent, or a general pattern. If the data analyzed correctly, this paper can explicitly distinguish the causal effect caused by habitat loss and fragmentation on species diversity/functional composition, a long-debated question since the proposed habitat amount hypothesis by Lenore Fahrig.

My main concern with this paper is that they should define the terms clearly and correctly and interpret their results carefully. For example, forest area can be an indicator of forest loss only when the original forest cover is the same. The effect of forest area on functional traits, such as decreasing wood density and seed mass, as well as the additional effect of local degradation, can also be interpreted as selective logging on hard-wood, late successional trees. To understand the observed pattern clearly, the authors may need to have a better understanding on the dynamics of forest landscapes, and consider the potential linkage between forest area (and loss), fragmentation, local degradation, and edge density: they are not independent factors. Please see my general and minor comments below, I hope they are useful to the improvement of this manuscript.

1. Definition of forest loss and fragmentation.

Forest loss was often defined as the loss of forest cover in a given period. In this paper, the authors used the inverse of forest cover as forest loss and assumed that all forest landscapes have an original cover of 100%, which is not reasonable in this case. The current forest cover could be the result of forest loss or natural expansion. It is the effect of habitat amount, but not the effect of loss. Therefore, the studied period is important and the authors should consider if there are available maps for decades ago, and calculate forest loss based on changes in forest cover.

Fragmentation is defined as the number of forest fragments in this study. However, habitat fragmentation is again a dynamic process and it is reasonable to consider whether this parameter (number of fragments) is linked to actual forest fragmentation process in these regions (e.g., a landscape with extremely high numbers of fragments can have lower numbers over time, leading to lower fragmentation). In addition, the effect of habitat fragmentation depends on time scales, and can persist for over a century (Vellend et al. 2006), so called "extinction debt", please make sure the time period of forest loss and fragmentation is considered.

2. The relationship between forest loss (cover), fragmentation, edge density and local degradation.

Local degradation was based on the inverse of tree community basal area. However, basal area is also strongly influenced by forest cover, fragmentation, edge density, as found by many of previous work (such as those by Brinck et al. 2017; Morreale et al. 2021; Pütz et al. 2014; Razafindratsima et al. 2017), and lots of work by Bill Laurance. Please check whether these factors are independent from each other, and whether basal area can be driven purely by local degradation.

3. Effect of spatial scales

The effect of spatial scales was reported for all figures, but it seems there was no major changes in results, except in a few cases such as in Fig. 3B, and the significant effect of fragmentation was only observed at smaller spatial scales in Paraiba. This is possibly the reason why the authors did not mention these results too much and there was no Discussion on it neither. To make the results clearer to readers, the authors may consider focusing on only one spatial scale (e.g., 1000 m) and mention that you have done the same analyses for other spatial scales in the supplementary files and you found similar results. However, if you keep these results in the main text, then you need to discuss a bit on the effect of these spatial scales.

4. The effect on large trees

This study used large trees $DBH > 10$ cm, which can grow decades or hundreds of years and then fragmentation effect may not be strong for them. However, edge effect and fragmentation may have strong impacts on young generations, especially that of saplings and young trees < 5 cm. This needs to be kept in mind that the observed pattern fits for large remaining trees in the fragments, and the effect for the whole community (e.g., trees $DBH > 1$ cm) is different.

Minor comments:

Line 79-80, Lewis et al 2015 mostly discuss the impacts of humans on tropical rainforest, but not on the importance of tropical rainforest to biodiversity and ecosystem services. Maybe consider cite (Pillay et al. 2022).

Line 81, please specify which period for this rate of deforestation.

Line 133-134, quantification of disturbance (using a proxy based on basal area), is this method reliable? How to clearly tell the difference between forest loss and fragmented induced reduced biomass, and those by other types of disturbance such as fire and selective logging?

Line 158, what is species niche centroids (SNC) and what it represents? This is rather unclear and should be clearer why it is linked to winner and loser species, because actually you have no data on winners and losers.

Line 408, the time period of forest loss and fragmentation matters. Loss and fragmentation are a dynamic process.

Discussion:

The discussion on winner and loser species is weak.

Why forest cover (loss) has a dominant role in driving functional trait changes is still not fully discussed, what is the mechanism?

Methods:

Line 539: Vegetation data: all trees are those > 10 cm in DBH, so this is more of an effect on large trees, not for young generations.

Line 515-544: Details about the plot size, number of plots, number of trees in each plot are very important, and Table S1 is very useful for readers to understand the details on distinct histories and intensities of land use for these six regions, please move it in the main text.

Trait: how many species have full trait data?

Line 584: the reasons to use 500-, 1000- and 2000 meter as buffer zones need to be told.

Line 655, definition of winners and losers, not very clear. Please state why this method is reliable.

References

- Brinck, K., Fischer, R., Groeneveld, J., Lehmann, S., Paula, M.D.D., Pütz, S., et al. (2017). High resolution analysis of tropical forest fragmentation and its impact on the global carbon cycle. *Nature communications*, 8, 14855.
- Morreale, L.L., Thompson, J.R., Tang, X., Reinmann, A.B. & Hutya, L.R. (2021). Elevated growth and biomass along temperate forest edges. *Nat Commun*, 12, 7181.
- Pillay, R., Venter, M., Aragon-Osejo, J., González-del-Piiego, P., Hansen, A.J., Watson, J.E.M., et al. (2022). Tropical forests are home to over half of the world's vertebrate species. *Frontiers in Ecology and the Environment*, 20, 10–15.
- Pütz, S., Groeneveld, J., Henle, K., Knogge, C., Martensen, A.C., Metz, M., et al. (2014). Long-term carbon loss in fragmented Neotropical forests. *Nature Communications*, 5, 5037.
- Razafindratsima, O.H., Brown, K.A., Carvalho, F., Johnson, S.E., Wright, P.C. & Dunham, A.E. (2017). Edge effects on components of diversity and above-ground biomass in a tropical rainforest. *Journal of Applied Ecology*, 38, 42–49.
- Vellend, M., Verheyen, K., Jacquemyn, H., Kolb, A., Calster, H.V., Peterken, G., et al. (2006). Extinction debt of forest plants persists for more than a century following habitat fragmentation. *Ecology*, 87, 542–548.

Reviewer #3 (Remarks to the Author):

I am only reviewing the structural causal model framework section for the manuscript "Winner-loser plant trait replacements in human-modified tropical forests". My main concern is that the DAG presented in Figure 1 consists only of the analyzed anthropogenic drivers and outcome variable of interest. However, for a DAG to be 'complete', all common cause variables (i.e., any variable that affects two or more variables already in the DAG) should also be included. It is possible that there are no other common cause variables required to depict the overall process, but I find this hard to believe. It would also be beneficial to test DAG-data consistency, as this can provide additional support to the presented DAG.

Version 1:

Decision Letter:

Dear Dr. Pinho,

Thank you for submitting your revised manuscript "Winner-loser plant trait replacements in human-modified tropical forests" (NATECOLEVOL-24041004A). It has now been seen again by the original reviewers, whose comments are below. Based on the reviewers' feedback, we will be happy in principle to publish it in *Nature Ecology & Evolution*, pending minor revisions to satisfy the reviewers' final requests and to comply with our editorial and formatting guidelines.

Thank you again for your interest in *Nature Ecology & Evolution*. Please do not hesitate to contact me if you have any questions.

[redacted]

Reviewer #1 (Remarks to the Author):

I appreciate the thoughtful responses to the reviewer's feedback and corresponding changes in the manuscript. In particular, the addition of text to help readers interpret how observed effects of landscape metrics relate to the likely ecological mechanisms/drivers is helpful. On one hand, I feel that the paper would be stronger if it presented a still more detailed framework and discussion focused on how these analyses of landscape metrics can disentangle which ecological mechanisms/processes are most/least important for causing the winner-loser spectrum. But on the other hand, I also understand that this focus on ecological mechanisms may be a longer-term goal beyond the ongoing debates focused on landscape metrics, and so I feel that this analysis can offer a useful advance for the field.

Reviewer #2 (Remarks to the Author):

I have read the response letter and the revised parts of the manuscript, and feel the authors have addressed my concerns well. The paper is in pretty good shape and I am happy to see it published in the journal. One thing I want to mention is that it seems my earlier comments are too straightforward and not that friendly, I am sorry for this but hope they are helpful to the authors.

Reviewer #3 (Remarks to the Author):

The authors did a great job at clarifying their assumptions within the SCM workflow. No further edits required on my end.

Version 2:

Decision Letter:

30th October 2024

Dear Dr Pinho,

We are pleased to inform you that your Article entitled "Winner-loser plant trait replacements in human-modified tropical forests", has now been accepted for publication in *Nature Ecology & Evolution*.

Over the next few weeks, your paper will be copyedited to ensure that it conforms to *Nature Ecology and Evolution* style. Once your paper is typeset, you will receive an email with a link to choose the appropriate publishing options for your paper and our Author Services team will be in touch regarding any additional information that may be required

Due to the importance of these deadlines, we ask you please us know now whether you will be difficult to contact over the next month. If this is the case, we ask you provide us with the contact information (email, phone and fax) of someone who will be able to check the proofs on your behalf, and who will be available to address any last-minute problems . Once your paper has been scheduled for online publication, the Nature press office will be in touch to confirm the details.

Acceptance of your manuscript is conditional on all authors' agreement with our publication policies (see www.nature.com/authors/policies/index.html). In particular your manuscript must not be published elsewhere and there must be no announcement of the work to any media outlet until the publication date (the day on which it is uploaded onto our web site).

Please note that *Nature Ecology & Evolution* is a Transformative Journal (TJ). Authors may publish their research with us through the traditional subscription access route or make their paper immediately open access through payment of an article-processing charge (APC). Authors will not be required to make a final decision about access to their article until it has been accepted. [Find out more about Transformative Journals](https://www.springernature.com/gp/open-research/transformative-journals)

Authors may need to take specific actions to achieve [compliance](https://www.springernature.com/gp/open-research/funding/policy-compliance-faqs) with funder and institutional open access mandates. If your research is supported by a funder that requires immediate open access (e.g. according to [Plan S principles](https://www.springernature.com/gp/open-research/plan-s-compliance)) then you should select the gold OA route, and we will direct you to the compliant route where possible. For authors selecting the subscription publication route, the journal's standard licensing terms will need to be accepted, including [those licensing terms](https://www.nature.com/nature-portfolio/editorial-policies/self-archiving-and-license-to-publish) will supersede any other terms that the author or any third party may assert apply to any version of the manuscript.

We welcome the submission of potential cover material (including a short caption of around 40 words) related to your

manuscript; suggestions should be sent to Nature Ecology & Evolution as electronic files (the image should be 300 dpi at 210 x 297 mm in either TIFF or JPEG format). Please note that such pictures should be selected more for their aesthetic appeal than for their scientific content, and that colour images work better than black and white or grayscale images. Please do not try to design a cover with the Nature Ecology & Evolution logo etc., and please do not submit composites of images related to your work. I am sure you will understand that we cannot make any promise as to whether any of your suggestions might be selected for the cover of the journal.

You can generate the link yourself when you receive your article DOI by entering it here: <http://authors.springernature.com/share>.

[redacted]

P.S. Click on the following link if you would like to recommend Nature Ecology & Evolution to your librarian
<http://www.nature.com/subscriptions/recommend.html#forms>

** Visit the Springer Nature Editorial and Publishing website at http://editorial-jobs.springernature.com?utm_source=ejP_NEcoE_email&utm_medium=ejP_NEcoE_email&utm_campaign=ejp_NEcoE for more information about our career opportunities. If you have any questions please click [here](mailto:editorial.publishing.jobs@springernature.com).

Reviewers' comments:

Reviewer #1 (Remarks to the Author):

This study examines the winner-loser trait spectrum under forest loss the Amazon and Atlantic forests and seeks to determine which trait changes are associated with several metrics/dimensions of anthropogenic forest modification. The study's major strength is that it presents a data synthesis large enough to explore whether winner-loser trait changes are consistent across this broad region. The analysis is detailed and interpreted carefully. The results show quite consistent impacts of forest loss on a subset of the traits measured, but no particularly strong and consistent causal effects of three descriptors of forest landscape change. Apart from a few specific comments, my one larger comment is that the authors could make more of the detailed results to help disentangle which ecological drivers (rather than landscape metrics) cause the winner-loser spectrum.

The anthropogenic drivers that the study focuses on (fragmentation, edge density, local degradation) are not really ecological processes that directly alter plant functional trait distributions. The paper does mention several ecological processes that have direct demographic effects on plants: biophysical edge effects (the authors discuss wind/storm damage, but heat/desiccation are relevant too), fire, and alteration of seed dispersal. Although I don't expect this study to incorporate data on these processes, I think the Discussion section could better leverage the results to disentangle which of these ecological drivers are stronger / weaker mechanistic explanations for the observed patterns. This could help orient future research on these direct ecological drivers causing this winner-loser spectrum.

R: We really appreciated the comments from Reviewer#1, which helped us to delve deeper into the important discussion of the ecological processes underlying the observed landscape-scale disturbance effects on local community functional profiles.

Firstly, we improved and extended the discussion on potential mechanisms behind the prevalent effects of forest loss on community functional profiles – see lines 239-246:

“The strong overarching influence of forest loss on the functional profiles of tree communities supports our expectation that part of the effects of other drivers are mechanistically associated with (and therefore can be predicted by) forest loss (Fig. 1a). For instance, the negative effects of forest loss often capture those of fragmentation and edge density, as increasing forest loss usually implies increased fragmentation, and therefore edge density^{8,32}. Beyond capturing landscape configuration effects, Fahrig's (2013) habitat amount hypothesis³³ suggests that the amount of habitat in surrounding landscapes is an important determinant of local species diversity, as more habitat equates to more individuals and species that could potentially colonize a local community. It remains to be seen how these potential mechanisms related to diversity could relate to functional profiles, but it is reasonable to expect that forest loss particularly limits the arrival of species with lower fecundity and dispersal capacity, which may explain the predominance of opportunistic species with high fecundity and dispersal capacity (small and endozoochoric seeds) in the more deforested landscapes we examined.”

Before delving into the mechanisms underlying the independent causal effects of the other drivers (see below), we have added some discussion on the inconsistent effects of edge density, which agree with the theory and empirical evidence to date, as explained in lines 255-260:

“[...] the inconsistent but often significant trait-level responses to edge density also require further research but may reflect the complexity of abiotic and biotic edge effects that are highly variable in space and time as they depend on multiple factors, such as matrix type, edge age and orientation, and interactions among nearby edges⁴⁰⁻⁴². Indeed, the “landscape-divergence hypothesis” suggests that edge effects are spatially variable and temporally dynamic⁴¹, which can result in contrasting successional trajectories in different regions⁴³”

For example, if the prediction at L422 that “trees with high densities can be especially vulnerable to stem breakage with extreme winds in edge-affected habitats” were true, then we should expect negative values for wood density primarily associated with Edge Density (Fig. 3C) and Fragmentation (Fig. 3B). Because the only significant values are instead positive, this ecological mechanism does not appear to be supported. Similarly, if leaf economics traits mediated the changes in edge/disturbed habitats, then we would probably expect traits like LMA to respond consistently to edge density and local degradation in particular, which also wasn’t observed. If the major driver were fire, which often affects forests near edges, then we might expect the largest significant effects across traits to be associated with edge density rather than just forest loss – again something not supported by these results.

R: We now provide more evidence and discussion on these mechanisms in lines 275-308, aligned with the answers made to previous comments. In particular, we agree that our previous interpretation of the observed effects of edge density could be improved. Therefore, in addition to the discussion on the inconsistent effects of edge density as shown above, we reformulated the related discussion in lines 285-295,

“Some results go against the prevailing understanding of trait responses to human disturbance. For example, we expected wood density to be negatively associated with edge density, but observed either no independent effect, or an increased dominance of hard-wooded species in response to edge density in some Amazonian regions. While fast-growing pioneer trees may benefit from increased light availability in forest edges^{13,15}, the harsh conditions of desiccated soil and air (i.e. water stress)⁴⁰, frequent fires and sporadic windstorms, could also favour hard-wooded trees with conservative resource-use traits that make them tolerant to abiotic stresses and resistant to stem snapping and uprooting^{47,48}. This finding highlights the context-dependency of responses of tropical flora to disturbance regimes, and demonstrates the need for more research that can understand and unpick the role of land-use history and biogeography^{20,49}, including the use of physiological traits that provide direct mechanistic links between environmental drivers and plant responses⁵⁰.”

We must also note that, given the assumptions of our conceptual model and consequently the structure of the models used to estimate the causal effects of each driver, as illustrated by the DAG in Fig. 1 and its legend, summarized in the introduction (e.g. lines 117-124) and detailed in the “Data analysis” section (e.g. lines 449-483), the effects of fragmentation and edge density may be captured by the total effects of forest loss and the inevitable fragmentation and edge creation this involves, as discussed in lines 235-239,

“The strong overarching influence of forest loss on the functional profiles of tree communities supports our a priori expectation that part of the effects of other drivers are mechanistically associated with (and therefore can be predicted by) forest loss (Fig. 1a). For instance, the negative effects of forest loss often capture those of fragmentation and edge density, as increasing forest loss

usually implies increased fragmentation, and therefore edge density (Villard & Metzger 2014, Fischer et al. 2021)."

This means that fragmentation and edge density would only have a significant effect in our models if their effects were additional to the total effect of forest loss – The comments from R1 and R2 demonstrated that this can be conceptually challenging, and that we did not do enough to clarify it in the first version. We have therefore detailed the structure and implications of our causal framework in the lines referred above, and we now refer to “independent effects” of anthropogenic drivers throughout the manuscript. See below more about this topic in response to another question.

In contrast to some of the other drivers that should be most strongly associated with edge density etc, I expect loss of large-bodied seed dispersers to be primarily determined by habitat loss (forest loss) and to a lesser extent fragmentation, and thus expect their effects to be borne out in panel A, which is consistent with the results. Also because dispersal traits and traits linked to dispersal by large-bodied seed dispersers (large-seeded, high density woods) are the traits most consistently associated with the winner-loser spectrum, it appears that loss of large dispersers in human-modified tropical forests is the best supported ecological driver of the winner-loser spectrum.

R: Indeed, we agree that defaunation and related dispersal limitation should be one of the main ecological processes behind the observed effects of forest loss. We have reformulated the related discussion in lines 278-284,

“[...] One of the most pervasive changes is the loss of large-seeded tree species, whose combined dependence on large-bodied seed dispersers and physiological requirements for germination make them especially vulnerable to both local and landscape-scale disturbances^{24,26,45}. Although the association between seed size and wood density is weak at the species level, it can be strongly expressed at the community level due to the dominance of certain small-seeded and low-wood density species dispersed by the many disturbance-adapted and highly mobile bats and birds that proliferate in human-modified landscapes^{24,26,46}.”

The authors would likely make other/different interpretations, but including in the Discussion and perhaps in the Abstract some reflections along these lines would help this paper make a larger advance, including by potentially helping move this field past the focus on disentangling the effects of different measures of landscape modification (number of patches vs edge density), which are analyzed as potential proxies of ecological drivers, and toward a better understanding of which direct ecological drivers are most important and of their functional effects in human-modified forest landscapes.

R: As detailed above, we have followed the suggestions from R1 and discussed the ecological mechanisms that are likely to underly the observed effects of landscape modification, and how further research could help moving towards a more mechanistic understanding of functional changes in human-modified landscapes. We think this has substantially improved the manuscript and we hope you agree.

We also would like to highlight that the challenge of disentangling the effects of different landscape metrics is still an open and hot debate, as evidenced by recent publications in top journals (e.g.,

Chase et al. 2020, Riva & Fahrig 2023, Zhang & Chase 2024). Therefore, regardless of whether the ecological mechanisms are clear or not, we believe that our paper makes large advances on this debate by (1) using a large amount of floristic and trait data including six regions, (2) applying a causal inference framework, (3) addressing mechanisms behind observed changes in functional composition (e.g., winner-loser replacements), and (4) using novel and powerful approaches like dc-CA, which account for potential issues in CWM regressions (Peres-Neto et al. 2017).

Specific comments

L224. At the risk of adding further figures, I'm unsure why only the upper 95% percentile was considered and not a lower percentile as well in this case. For example, an increasing prevalence of very small-seeded / weedy species is potentially equally meaningful as change in the largest seed sizes, and while that's reflected in some of the other analysis, it isn't shown here.

R: This is an interesting point. We initially chose to assess the 95% percentile as there are strong a priori reasons why large seeds face a dispersal bottleneck (see citations and discussions in other responses) and these have disproportional importance for ecosystem functioning, as mentioned in the introduction (lines 92-94). Furthermore, we did not want to add more analysis as (1) to avoid too much complexity as the manuscript already cover many regions, traits, spatial scales, anthropogenic drivers and response variables, and (2) because we found that most of these changes in other trait moments are associated with the observed changes in CWMs (Fig. 4).

Nonetheless, as suggested we have performed the same analysis for the 5% lower percentiles of trait distributions and included the related graphics as Extended Data Figs. 3 and 4. It interestingly shows that there are also frequent negative effects of forest loss and local degradation on this attribute. As discussed in lines 322-326,

"[...] This suggests that these drivers usually push the whole trait distributions toward lower values – i.e., increasing dominance of opportunistic strategies but also limiting extremely conservative resource-use strategies and allowing the presence of extremely acquisitive strategies that are absent from more conserved landscapes"

Figure 3. It would help readers to provide a description of the ($\sim A | B$) notation within the figure legend. Or in the text walk readers through the relationship to the causal links drawn in Fig 1 (ie in order to assess local degradation, the analysis needs to account for the effects of forest loss, fragmentation, and edge density). Figure 4's caption provides more explanation of this than does Figure 3.

R: As suggested, we clarified the meaning of this notation in the legend of Figure 3.

There are some cited papers missing from the References list.

R: Thanks, we cross-checked the citations and reference list and added missing refs.

Reviewer #2 (Remarks to the Author):

This study explored the independent effects of forest loss, fragmentation, and degradation on plant functional composition in tropical rainforests with distinct histories and intensities of land use. They conclude that forest loss has a dominant role in driving functional trait composition, and local degradation can amplify the effects of forest loss. However, against most case studies that did not

distinguish the effect of fragmentation per se, they found that the effect of landscape configuration on functional trait composition is rare.

This interesting study has a large dataset involving forest landscapes (area, number of fragments, and edge density), species composition, community structure (basal area) and functional traits from 271 forest plots in six regions of Amazon. Therefore, the results from this study can make great contributions to test whether previous case studies on this topic (e.g., Magnago et al. 2014) is context-dependent, or a general pattern. If the data analyzed correctly, this paper can explicitly distinguish the causal effect caused by habitat loss and fragmentation on species diversity/functional composition, a long-debated question since the proposed habitat amount hypothesis by Lenore Fahrig.

R: Thank you, we are glad to know that you found our paper interesting and acknowledge its potential to advance the field.

My main concern with this paper is that they should define the terms clearly and correctly and interpret their results carefully. For example, forest area can be an indicator of forest loss only when the original forest cover is the same. The effect of forest area on functional traits, such as decreasing wood density and seed mass, as well as the additional effect of local degradation, can also be interpreted as selective logging on hard-wood, late successional trees. To understand the observed pattern clearly, the authors may need to have a better understanding on the dynamics of forest landscapes, and consider the potential linkage between forest area (and loss), fragmentation, local degradation, and edge density: they are not independent factors. Please see my general and minor comments below, I hope they are useful to the improvement of this manuscript.

R: We appreciate the comments from R2, which helped us to carefully think and improve or justify the terms adopted and to clarify the structure and implications of our causal inference framework, as detailed below.

1. Definition of forest loss and fragmentation.

Forest loss was often defined as the loss of forest cover in a given period. In this paper, the authors used the inverse of forest cover as forest loss and assumed that all forest landscapes have an original cover of 100%, which is not reasonable in this case. The current forest cover could be the result of forest loss or natural expansion. It is the effect of habitat amount, but not the effect of loss. Therefore, the studied period is important and the authors should consider if there are available maps for decades ago, and calculate forest loss based on changes in forest cover.

R: We respond to this in two ways – (1) about the reasonableness of our assumption that all regions had 100% forest cover, and (2) about the reasons for adopting this terminology.

(1) First, for the assumptions of forest cover, it is important to remember that our measures of forest loss are in buffers extending a maximum of 2km around each sampling plot – with most analysis showing similar results for buffers of just 500m; we are not extrapolating to large areas where you would find non-forest formations within the forested regions, we are using local assessments. Furthermore, we have good reasons to believe that, at least in a very long time scale, all non-water parts of the study regions were completely covered by forest and therefore the inverse of current forest cover would represent forest loss at the long-term.

For example, in the Amazon:

- The Balbina dam was completely forested, and the region was only filled from 1987 https://philip.inpa.gov.br/publ_livres/preprints/1989/balbina-eng2.pdf
- Forest loss in Paragominas only began with the construction of the Belem-Brasilia highway and founding of the city in 1961 <https://paragominas.pa.gov.br/o-municipio/historia/>
- While Santarem has a longer history of colonization and has some natural savanna enclaves, the geographical distribution of these is distinct from the forest formations. The data we used follows the sampling design outlined in Gardner et al. 2013, which focused explicitly on catchments lying entirely within the naturally forested zone.
- Large-scale deforestation occurred in the Atlantic Forest biome much longer ago than the Amazon, but all plots we assess are within the estimated limits of the original extent (Ribeiro et al. 2009, Vancine et al. 2024). Furthermore, the extent of forest can to some extent be inferred by the agricultural activities that replaced the forests. In Serra Grande and Paraíba, the main agricultural activity that replaced forests – sugar cane – is focused explicitly on the regions of fertile soils where forests would have been. The sites in Bahia are all within the forested zone – essential for maintaining the Cabruca system of cocoa farming that is one of the main land uses. In Paraiba, the matrix is an urban environment which would have originally been coastal forest.

We now explain this in the introduction – see lines 124-126:

“As all regions were originally covered by tropical forests, we measured forest loss as the percentage of the landscape covered by non-forest land covers, including cattle pastures, agricultural lands, and human settlements. This percentage thus represents the cumulative forest loss to date.”

Also, we now explicitly refer to it as *“The percentage of forest loss in the surrounding landscapes”* in the Results (e.g., line 150) and Methods (e.g. line 194), to make clear that our forest loss measure does not account for temporal dynamics – the implication of this is also discussed in lines 268-272 as shown below in response to the next question.

(2) Regarding the reasons for adopting this terminology, we prefer to use “forest loss” rather than “habitat amount” to facilitate the interpretation of our expectations and results, as introduced in Fig. 1B and explained in the Data Analysis section – see lines 488-493,

“Note that for all continuous functional traits (i.e., excluding dispersal syndromes), low values reflect opportunistic strategies and high values more conservative traits (e.g., we consider LMA instead of its inverse, SLA, since higher values means denser leaves). Similarly, all predictors considered are disturbance variables with values increasing with increasing level of disturbance, e.g. forest loss instead of forest cover. Therefore, we expected negative relationships for all these trait-driver combinations (Fig. 1B)”.

Fragmentation is defined as the number of forest fragments in this study. However, habitat fragmentation is again a dynamic process and it is reasonable to consider whether this parameter (number of fragments) is linked to actual forest fragmentation process in these regions (e.g., a landscape with extremely high numbers of fragments can have lower numbers over time, leading to lower fragmentation). In addition, the effect of habitat fragmentation depends on time scales, and can persist for over a century (Vellend et al. 2006), so called “extinction debt”, please make sure the time period of forest loss and fragmentation is considered.

R: We agree that fragmentation is a dynamic process, but we used the number of fragments at a single point in time as a standardized landscape metric to capture the landscape configuration at a similar point in time as when the communities were sampled. This is the prevailing approach in most studies in landscape ecology (e.g., Fahrig 2013, Watling et al. 2020, Arasa-Gisbert 2022).

We note that the precaution of the reviewer comes from a widespread confusion on the meaning of fragmentation as a process, as mentioned by the reviewer, and fragmentation as a pattern, the approach we used and that is widely adopted in the literature.

Also, although some of the study regions, in particular Paragominas, have experienced some changes in the decades prior to sampling, exploring the time lags and “functional” extinction debts is beyond the scope of this manuscript. Nonetheless, we raise this possibility at the end of the discussion as we agree this is an interesting avenue for further research that our paper could stimulate – see lines 268-272,

“[...] we were limited by relying on snapshots of tree communities at specific times in each region. Understanding the temporal dynamics of human modified forests will help reveal the time lags between disturbances and functional changes (i.e. the functional extinction debt) and how tree species and traits are responding to climate change and its interactions with local anthropogenic pressures.”

2. The relationship between forest loss (cover), fragmentation, edge density and local degradation. Local degradation was based on the inverse of tree community basal area. However, basal area is also strongly influenced by forest cover, fragmentation, edge density, as found by many of previous work (such as those by Brinck et al. 2017; Morreale et al. 2021; Pütz et al. 2014; Razafindratsima et al. 2017), and lots of work by Bill Laurance. Please check whether these factors are independent from each other, and whether basal area can be driven purely by local degradation.

R: R2 is right to say that the anthropogenic drivers we have analyzed are not independent – they are definitely not. We clearly acknowledge this by adopting a causal inference framework, in which we define our DAG (Fig. 1) and structure our models accordingly to estimate the total and independent causal effects of each driver by accounting for the expected relationships among them. In the specific case of local degradation (i.e., the inverse of community basal area), the observed effects are controlled for all landscape drivers, so that it should be driven by local disturbances, as explained in the introduction (lines 119-123), Fig. 1 legend (lines 777-781) and with more detail in the Methods (see lines 475-480). Also, as detailed below in response to the comment from R3, other potential drivers of community basal area occur at larger scales (e.g., climate) and are also controlled in our within-region approach as explained in lines 453-465).

The need to advance understanding of the *independent* (or causal) effects of the analyzed drivers was outlined in the beginning of the introduction (lines 58-61),

“While there is broad consensus that habitat loss is one of the main causes of the contemporary biodiversity crisis⁵⁻⁷, the independent effects of landscape configuration – here referring to both the number of forest patches and edge density – or local disturbances leading to degradation, remain debated⁸⁻¹⁰ or are poorly studied⁴.”

We also have provided an overview of our approach to this problem in the end of the Introduction, which has now been reformulated and extended – see lines 117-123,

“We adopt a structural causal modelling framework^{30,31} to estimate the total causal effects of each disturbance variable according to our assumptions of causal relationships among them and with the outcome variables (Fig. 1). In this framework, we assume from well-documented relationships that forest loss leads to increased number of forest patches (i.e. fragmentation), both of which increase forest edge density^{8,32}, and that these three landscape-scale drivers lead to increased local degradation⁴. To account for these multiple cause-effect relations, we used separate models with different sets of control variables aimed at estimating the causal effects of each driver on the outcomes of interest (Fig. 1, details in Methods).”

This is further detailed in the Data Analysis section - see lines 449-480 for a complete description of our causal framework.

Finally, we also discuss the implications and limitations of our causal framework in lines 261-272,

“The use of a causal framework allowed us to infer total causal effects of our disturbance variables while minimizing risks of overcontrol, confounding or collider biases^{30,31}. However, the inferences remain conditional on the accuracy of the causal model (Fig. 1) and the presence and frequency of different condition combinations in the available data (e.g., low forest loss, high fragmentation). Forest loss is an economic and social process that is likely to limit the empirical data to a subset of potential combinations. Going beyond this will require experimental landscapes (e.g., SAFE, www.safeproject.net) or the use of mechanistic simulations (e.g.,⁴⁴) that allow wider and orthogonal gradients of forest loss and fragmentation [...]”

3. Effect of spatial scales

The effect of spatial scales was reported for all figures, but it seems there was no major changes in results, except in a few cases such as in Fig. 3B, and the significant effect of fragmentation was only observed at smaller spatial scales in Paraíba. This is possibly the reason why the authors did not mention these results too much and there was no Discussion on it neither. To make the results clearer to readers, the authors may consider focusing on only one spatial scale (e.g., 1000 m) and mention that you have done the same analyses for other spatial scales in the supplementary files and you found similar results. However, if you keep these results in the main text, then you need to discuss a bit on the effect of these spatial scales.

R: First, as now reformulated in the Methods (lines 435-439):

“The selected scales encompass and extend beyond those used in other studies of landscape-scale disturbance effects on tree assemblages (e.g.²⁵). We adopted this broad multi-scale approach to assess the consistency of effects across scales as (i) there was no a priori information allowing us to select a single scale, and (ii) it was highly unlikely that a common “scale of effect”³³ would emerge across regions, traits, predictors, and response variables.”

Although overall we agree with R2 that the results were mostly consistent across scales, there were a number of cases in which the effects of a driver for a given trait-region were consistent in direction across scales but statistically significant only at one or two scales (see Fig. 3). Selecting one single scale would therefore lead to different conclusions depending on which one we select, when it only

reflects the context/trait dependence of the scale of effects as explained in the sentence above. For example, if we selected the largest scale (2,000 meter-radius buffer), which overall was the one capturing strongest forest-loss effects, this would lead to the conclusion that forest loss and fragmentation are not relevant in Balbina and Paraiba, respectively, when they are but at different (smaller) scales. These are only two examples, but there are several, which preclude us to simply select one single scale.

We have therefore decided to keep showing the results for all the analyzed scales in the graphics, as we believe that it does not hinder the visualization and interpretation of our results, while it shows the consistent direction but different magnitude of effects at different scales across trait-region-driver combinations.

4. The effect on large trees

This study used large trees DBH>10 cm, which can grow decades or hundreds of years and then fragmentation effect may not be strong for them. However, edge effect and fragmentation may have strong impacts on young generations, especially that of saplings and young trees <5 cm. This needs to be kept in mind that the observed pattern fits for large remaining trees in the fragments, and the effect for the whole community (e.g., trees DBH> 1cm) is different.

R: We agree that the effects could be different if including seedlings and saplings, but we focused on large (adult) trees because:

1) Saplings are much more abundant than adult trees and show strong differential mortality rates depending on key traits (Wright et al. 2010), therefore not reflecting the balance of the community that will become adult trees.

2) Focusing on big stems is therefore more reliable, and also potentially more conservative, as now mentioned in lines 392-394,

“The selection of adult trees is a conservative approach as landscape modification effects may be stronger and should manifest earlier in seedlings and saplings^{22,25}”

3) The approach we took is consistent with most studies in the related literature (e.g., Santos et al. 2008, Poorter et al. 2017, Chase et al. 2020), and therefore allows for potential comparisons.

4) Data on the saplings is only available for a subset of the regions, and including this would add significant complexity to the analysis and interpretation.

Minor comments:

Line 79-80, Lewis et al 2015 mostly discuss the impacts of humans on tropical rainforest, but not on the importance of tropical rainforest to biodiversity and ecosystem services. Maybe consider cite (Pillay et al. 2022).

R: We agree that Lewis et al 2015 was not the most appropriate reference to support the key role of tropical forests for biodiversity conservation and ecosystem services provisioning. We have therefore replaced it by Slik et al. 2015 (which shows the astonishing number of tropical tree species) and Brandon 2014 (which support their key role for ecosystem services provisioning). We think these are better alternatives than the one suggested by R2, which refers to vertebrates only.

Line 81, please specify which period for this rate of deforestation.

R: As required, we now specify that these deforestation rates have been observed over the last two decades (line 55).

Line 133-134, quantification of disturbance (using a proxy based on basal area), is this method reliable? How to clearly tell the difference between forest loss and fragmented induced reduced biomass, and those by other types of disturbance such as fire and selective logging?

R: Yes, we believe this is a reliable proxy, which has been adopted in many studies to track tropical year-on-year changes during forest succession (e.g. Gilman et al. 2016, Pinho et) or to evaluate the impacts of disturbances such as fires (e.g. Armenteras et al. 2021) – see also references in the main text. The difference between forest loss/fragmentation-induced basal area reduction and those imposed by local disturbances is explicitly considered in our causal inference framework, as explained above in response to R2's question about the independent effects of each driver and in particular of local degradation. Specifically, local degradation effects are controlled for all landscape drivers as explained in lines 119-124, Fig. 1 legend, and now with more detail in the methods, lines 449-480. This is also clear from the notation in Fig. 3 x-axis, which is now explained in its legend as required by R1.

Line 158, what is species niche centroids (SNC) and what it represents? This is rather unclear and should be clearer why it is linked to winner and loser species, because actually you have no data on winners and losers.

R: The definition of “species niche centroids” (the acronym has been removed to avoid confusion) was included in the Methods, in the “Species-level analyses” section (now with more detail in lines 512-525). To clarify it earlier, we now have included the definition in the Introduction as well, when referring to this term for the first time in lines 134-138,

“Species distributions along forest loss gradients were parameterized by species niche centroids, calculated as the abundance-weighted average forest loss across the plots where a species occurs. Winners and losers were defined as those species whose distribution differ from a random-dispersal null model³⁶, in which winners thrive with forest loss in opposition to losers whose abundance is reduced in more deforested landscapes”

Regarding the last comment from R2 here, we must note that actually no one has “data on winners and losers”, at least not in absolute terms, as there are many possible approaches to define it, depending on the study aim and system. See further discussion on this below, in response to R2 question about the reliability of our approach to define winners and losers.

Line 408, the time period of forest loss and fragmentation matters. Loss and fragmentation are a dynamic process.

R: We refer to our earlier response on this topic. In addition, it is important to note that we found broadly similar results for regions with very different times since forest loss, and very different matrices and disturbance regimes, etc. The six contexts are a real strength of our work and underpin the robustness of the generalities that emerge, as now emphasized in the abstract (lines 47-48) and

conclusion (350-352).

Discussion:

The discussion on winner and loser species is weak.

R: We respect the reviewer's opinion, but other reviewers do not see it in the same way and have highlighted the strength of our approach to address how the replacement of a few dominant winner-loser species can drive the observed changes in CWMs.

However, we have modified the related discussion in response to this and other comments – for instance, see lines 328-339,

“[...] we revisited the paradigm of the winner-loser species replacements^{14,19} and show how this can be useful to delve deeper into the interpretation of changes in CWM and trait-environment relationships across species²⁸. We found that although winners and losers were functionally distinct, they were not distinguished from species irresponsive to forest loss gradients, and both winners and losers covered large ranges of trait values (boxplots in Figs. 5 and S7). However, a few dominant winners and losers (large blue and red points in Fig. 5) were consistently found at the extremes of the traits' distribution, corresponding to either low or high trait values, respectively. This suggests that while many trait dimensions may ultimately determine the fate of species in human-modified landscapes, dominant winner and loser species have stronger and more consistent trait-environment associations – those embedded in landscapes with high forest cover consistently held extremely dense woods and large seeds dispersed by synzoochory (loser traits), while species dominant in highly deforested landscapes had low-density woods and small seeds dispersed by endozoochory (winner traits)”.

We would be glad to work on suggestions of other pathways to follow if these changes do not resolve the reviewer's criticism here.

Why forest cover (loss) has a dominant role in driving functional trait changes is still not fully discussed, what is the mechanism?

R: This is now better discussed with more detail in lines 235-246,

“The strong overarching influence of forest loss on the functional profiles of tree communities supports our expectation that part of the effects of other drivers are mechanistically associated with (and therefore can be predicted by) forest loss (Fig. 1a). For instance, the negative effects of forest loss often capture those of fragmentation and edge density, as increasing forest loss usually implies increased fragmentation, and therefore edge density^{8,32}. Beyond capturing landscape configuration effects, Fahrig's (2013) habitat amount hypothesis³³ suggests that habitat amount is an important determinant of local species diversity, as more available habitat equates to more individuals and species in the surrounding landscape that could potentially colonize a local community. It remains to be seen how these potential mechanisms related to diversity could relate to functional profiles, but it is reasonable to expect that forest loss particularly limits the arrival of species with lower fecundity and dispersal capacity, which may explain the predominance of opportunistic species with high fecundity and dispersal capacity (small and endozoochoric seeds) in the more deforested landscapes we examined.”

Methods:

Line 539: Vegetation data: all trees are those > 10 cm in DBH, so this is more of an effect on large trees, not for young generations.

R: We detailed the reasons and implications of selecting adult trees in response to a similar question above, and we included part of it in the Methods (lines 392-394).

Line 515-544: Details about the plot size, number of plots, number of trees in each plot are very important, and Table S1 is very useful for readers to understand the details on distinct histories and intensities of land use for these six regions, please move it in the main text.

R: Unfortunately, we must refrain from making this change, as we would exceed the maximum number of figures/tables, unless we removed some of the figures, which we consider more relevant than this table.

Trait: how many species have full trait data?

R: We included this information in lines 425-426,

“Over one third of the species (438/1,207) have species-level data for all analyzed traits.”

Line 584: the reasons to use 500-, 1000- and 2000 meter as buffer zones need to be told.

R: We now explain our reasons for selecting these scales in lines 435-439,

“The selected scales encompass and extend beyond those used in other studies of landscape-scale disturbance effects on tree assemblages (e.g.²⁵). We adopted this broad multi-scale approach to assess the consistency of effects across scales as (i) there was no a priori information allowing us to select a single scale, and (ii) it was highly unlikely that a common “scale of effect”³³ would emerge across regions, traits, predictors, and response variables.”

As the scale of effects should ultimately depend on the response variable and study region (San-José et al. 2019), we chose three scales that we believe would likely have an effect. In addition to 1,000 and 2,000 meters-radius buffers, which have been identified as the scale of effect in other studies, 500m is interesting as it captures the changes in immediate vicinity, including the extended tree mortality from edge effects that can extend up to 100 meters into forests (Laurance et al. 2018) or changes in vertebrate use of up to 400m (Pfeifer et al. 2017).

Line 655, definition of winners and losers, not very clear. Please state why this method is reliable.

R: We now summarize our approach to define winners and losers in the introduction as shown above (lines 134-138), and provide more details in the methods, including some discussion on the reliability of our approach – see lines 512-525:

“... we calculated species’ forest-loss niche centroid, defined as the average of forest loss across the plots where the species occurs, weighted by its abundance in each plot. We then applied a null model approach³⁶ in which we randomly distributed the abundance of species across plots within regions and re-calculated their niche centroid, we did so 10,000 times for each species-region combination. Standard effect sizes (SES) were then calculated for each species in each region and at each local

landscape scale, to describe the direction and magnitude of the deviation of observed niche centroid from random expectation, with positive and negative values reflecting species associated with more and less disturbed conditions, respectively. We considered species with observed niche centroid higher or lower than expected by chance (i.e. than 95% of the random values obtained from the 10,000 null-model iterations), as potential winners or losers, respectively. This method has been successfully applied to identify winners and losers of land-use intensification³⁶ and has the advantage of being based on abundance-weighted niche centroids, allowing species that have reduced abundance in deforested landscapes to be defined as losers and species that are usually present but thrive in disturbed sites as winners.”

Reviewer #3 (Remarks to the Author):

I am only reviewing the structural causal model framework section for the manuscript "Winner-loser plant trait replacements in human-modified tropical forests". My main concern is that the DAG presented in Figure 1 consists only of the analyzed anthropogenic drivers and outcome variable of interest. However, for a DAG to be 'complete', all common cause variables (i.e., any variable that affects two or more variables already in the DAG) should also be included. It is possible that there are no other common cause variables required to depict the overall process, but I find this hard to believe. It would also be beneficial to test DAG-data consistency, as this can provide additional support to the presented DAG.

To clarify, the DAG does not need to include all relevant variables to be complete. However, it does need to include all common cause variables. These are variables that affect (have an arrow pointing into) two or more variables already presented in the DAG. These must be included to account for confounding, and should be included whether they are measured or unmeasured variables (there are ways to deal with unmeasured confounding that the authors can employ in their analysis, for example). At present, it did not seem like the authors included all common cause variables in their DAG but rather just included their outcome and predictor variables of interest. If the DAG is not complete, however, then the analysis cannot claim to be causal.

R: R3 raises two important points related to structural causal modelling frameworks in general. Firstly, the need for causal diagram to go beyond measured variables – whenever relevant to the studied system –, meaning explicitly considering unobserved or latent variables, if any, in the causal diagram (DAG) to ensure that no forgotten confounding paths remain open (McElreath 2020). This is important when defining “adjustment sets” (sets of control variables aimed at closing backdoor paths) to quantify a causal estimate of interest while minimising risks of biases. Secondly, once a DAG is defined, based on expert knowledge and existing literature, R3 emphasises the good practice of testing the consistency between the DAG and the available data, as a way to validate its plausibility before using it to generate causal estimates relating to the questions of the work. This step is performed by exploring the testable implications derived from the DAG. These consist of conditional independencies, through which pairs of variables are expected to be independent (not associated) once we condition on the variable(s) necessary to *d*-separate them (Pearl 2009, Ankan et al. 2021).

Regarding potential missing variables in the DAG:

Reading R3’s comment about the absence of unmeasured but ecologically relevant variables in the

causal diagram, we realised that some additional explanation was needed in the Methods section of the study to clarify our workflow as well as working assumptions. These can now be found in lines 453-468 in the revised manuscript.

We did think of potential unobserved confounders early on, in the studied system, but did not find any that would act **within** each of the studied regions, i.e. a key point that we now clarify both in the Methods (lines 461-465). As such, not having latent variables in the DAG was a conscious decision, not an oversight. A key point to this is indeed that, as now included in Fig. 1 legend (lines 790-792),

“We performed our analyses separately for each of the six regions (Fig. 2), where forest plots are relatively homogeneous in terms of climate and soil types, to limit the risks of potential unobserved confounders of the represented cause-effect relations; this should therefore be read as a “within-region DAG.”

Also, as further discussed in the Methods - see lines 453-465:

“Causal thinking requires to account for both observed and unobserved but relevant variables or processes⁷⁵. Here, a potential unobserved confounder of the represented cause-effect relations (Fig. 1) could be average climate and soil types. For example, at broad spatial scales going across regions and marked gradients, wetter climates could increase forest productivity and therefore forest degradation, while independently controlling the functional assembly of tree communities (through spatial partitioning of species and life-history strategies relating to their fundamental and realized niches; e.g.⁷⁶). Such “common cause” structures in a DAG can lead to spurious associations between the variable whose effect we investigate (e.g., forest degradation) and the outcome (confounder bias)³¹. Here, to limit the risks of such biases, we performed the analysis of each causal effect within relatively homogeneous regions, climate- and soil-wise, to close potential non-causal paths of associations through such broad-scale unobserved variables. Our DAG does not represent this unobserved confounder structure, and is therefore to be read as a “within-region DAG”.

Therefore, we did not represent these climate, or soil-related variables, in the DAG precisely because our analyses are run *within* regions, which allows us to stratify by these variables and block associated non-causal paths of associations between anthropogenic pressure variables and the functional responses. The DAG therefore represents key relevant processes *at the spatial scale of the analysis*, without need for potential latent confounders acting at broader spatial scales to be presented.

On the good practice of testing DAG-data consistency:

R3 is right to emphasise that this step was missing in the initial submission. The reason why this did not appear was that our DAG does not imply any testable implications (but see next point on validation). Our mistake was not to mention it, and this is now corrected – see lines 466-468:

“The second step consisted of investigating the DAG-data consistency by testing the DAG’s testable implications, i.e., its conditional independencies³⁰. However, our particular DAG did not have any testable implications.”

This absence of conditional independencies can be verified directly on dagitty.net, copying the following ‘model code’ (to replicate our DAG) and then checking the ‘Testable Implications’ box on the right hand-side, or in R, with the following:

```

library(dagitty)
myDAG <- dag {
  LocDeg -> WD
  e_dens2000 -> LocDeg
  e_dens2000 -> WD
  lnFL2000 -> LocDeg
  lnFL2000 -> e_dens2000
  lnFL2000 -> lnNfrag2000
  lnNfrag2000 -> LocDeg
  lnNfrag2000 -> WD
  lnNfrag2000 -> e_dens2000
}
plot(myDAG) # Visualise the DAG
impliedConditionalIndependencies(myDAG) # generate testable implications of the DAG

```

where WD: wood density, lnFL2000: ln of forest loss, lnNfrag2000: ln of number of fragments, e_dens2000: edge density, and LocDeg: local degradation.

The output of this last line of code confirms there are no testable implications in our DAG.

We believe our additions and clarifications add robustness to the analytical decisions we made while better acknowledging assumptions upon which our inference relies.

Additional validation of the DAG:

A final important point further supporting our inference comes from the confirmatory results obtained through the double constrained correspondence analysis (dc-CA), a complementary multivariate analytical method that provided support to the results of the causally explicit analyses above. Indeed, the dc-CA analysed all abundance, trait and environment data jointly, and reached the same results independently of the DAG: the 2000m scale forest loss is the dominant driver, followed by local degradation (with or without adjustment for the other landscape metrics). We now mention how this second analytical approach provides support to the first one in the Results (line 213 and line 221) and further discuss it in lines 318-320,

“[...] A dc-CA with forward selection of disturbance variables at the three spatial scales led to the same final model as the DAG-based dc-CA modelling approach, but would have had less power and would not have allowed causal inference.”

References

- Ankan, A., Wortel, I. M. N. and Textor, J. 2021. Testing Graphical Causal Models Using the R Package “dagitty.” - *Curr. Protoc.* 1: e45.
- Arasa-Gisbert, R., Arroyo-Rodríguez, V., Meave, J. A., Martínez-Ramos, M. & Lohbeck, M. Forest loss and treeless matrices cause the functional impoverishment of sapling communities in old-growth forest patches across tropical regions. *J. App. Ecol.* 59, 1897–1910 (2022).
- Armenteras, D. *et al.* Fire threatens the diversity and structure of tropical gallery forests. *Ecosphere* **12**, e03347 (2021).
- Chase, J. M., Blowes, S. A., Knight, T. M., Gerstner, K. & May, F. Ecosystem decay exacerbates biodiversity loss with habitat loss. *Nature* 584, 238–243 (2020).

- Fahrig, L. Rethinking patch size and isolation effects: the habitat amount hypothesis. *J. Biogeog.* 40, 1649–1663 (2013).
- Gilman, A. C. *et al.* Recovery of floristic diversity and basal area in natural forest regeneration and planted plots in a Costa Rican wet forest. *Biotropica* 48, 798–808 (2016).
- Laurance, W. F. *et al.* An Amazonian rainforest and its fragments as a laboratory of global change. *Biol. Rev.* 93, 223–247 (2018).
- McElreath, R. 2020. *Statistical Rethinking (Second Edition): A Bayesian Course with Examples in R and Stan.* - CRC Press.
- Pearl, J. *Causality: Models, reasoning and inference.* 2nd ed. (Cambridge University Press, 2009).
- Peres-Neto, P. R., Dray, S. & Ter Braak, C. J. F. Linking trait variation to the environment: critical issues with community-weighted mean correlation resolved by the fourth-corner approach. *Ecography* 40, 806–816 (2017).
- Pfeifer, M. *et al.* Creation of forest edges has a global impact on forest vertebrates. *Nature* 551, 187–191 (2017).
- Poorter, L. *et al.* Biodiversity and climate determine the functioning of Neotropical forests. *Glob. Ecol. Biogeog.* 26, 1423–1434 (2017).
- Ribeiro, M. C., Metzger, J. P., Martensen, A. C., Ponzoni, F. J., & Hirota, M. M. (2009). The Brazilian Atlantic Forest: How much is left, and how is the remaining forest distributed? Implications for conservation. *Biological Conservation* 142(6), 1141-1153.
- Riva, F. & Fahrig, L. Landscape-scale habitat fragmentation is positively related to biodiversity, despite patch-scale ecosystem decay. *Ecol. Lett.* 26, 268–277 (2023).
- Santos, B. A. *et al.* Drastic erosion in functional attributes of tree assemblages in Atlantic forest fragments of northeastern Brazil. *Biol. Cons.* 141, 249–260 (2008).
- Vancine, M. H. *et al.* The Atlantic Forest of South America: Spatiotemporal dynamics of the vegetation and implications for conservation. *Biol. Cons.* 291, 110499 (2024).
- Villard, M. & Metzger, J. P. REVIEW: Beyond the fragmentation debate: a conceptual model to predict when habitat configuration really matters. *J. Appl. Ecol.* 51, 309–318 (2014).
- Watling, J. I. *et al.* Support for the habitat amount hypothesis from a global synthesis of species density studies. *Ecol. Lett.* 23, 674–681 (2020).
- Wright, S. J. *et al.* Functional traits and the growth–mortality trade-off in tropical trees. *Ecology* 91, 3664–3674 (2010).
- Zhang, H., Chase, J. M. & Liao, J. Habitat amount modulates biodiversity responses to fragmentation. *Nat Ecol Evol* 8, 1437–1447 (2024).